# Structural insight into molecular mechanism of poly (ethylene terephthalate) degradation

Seongjoon Joo[1], In Jin Cho [2], Hogyun Seo[1], Hyeoncheol Francis Son[1], Hye-Young Sagong[1], Tae Joo Shin[3], So Young Choi [2], Sang Yup Lee [2] & Kyung-Jin Kim[1]

Plastics, including poly(ethylene terephthalate) (PET), possess many desirable characteristics and thus are widely used in daily life. However, non-biodegradability, once thought to be an advantage offered by plastics, is causing major environmental problem. Recently, a PET-degrading bacterium, *Ideonella sakaiensis*, was identified and suggested for possible use in degradation and/or recycling of PET. However, the molecular mechanism of PET degradation is not known. Here we report the crystal structure of *I. sakaiensis* PETase (*Is*PETase) at 1.5 Å resolution. *Is*PETase has a Ser–His–Asp catalytic triad at its active site and contains an optimal substrate binding site to accommodate four monohydroxyethyl terephthalate (MHET) moieties of PET. Based on structural and site-directed mutagenesis experiments, the detailed process of PET degradation into MHET, terephthalic acid, and ethylene glycol is suggested. Moreover, other PETase candidates potentially having high PET-degrading activities are suggested based on phylogenetic tree analysis of 69 PETase-like proteins.

[1] School of Life Sciences (KNU Creative BioResearch Group), KNU Institute for Microorganisms, Kyungpook National University, Daehak-ro 80, Buk-gu, Daegu, 41566, Republic of Korea. [2] Metabolic and Biomolecular Engineering National Research Laboratory, Department of Chemical and Biomolecular Engineering (BK21 Plus Program), BioProcess Engineering Research Center, and KAIST Institute (KI) for the BioCentury, Korea Advanced Institute of Science and Technology (KAIST), 291 Daehak-ro, Yuseong-gu, Daejeon, 34141, Republic of Korea. [3] UNIST Central Research Facilities & School of Natural Science, Ulsan National Institute of Science and Technology, 50 UNIST-gil, Eonyang-eup, Ulju-gun, Ulsan, 44919, Republic of Korea. Seongjoon Joo, In Jin Cho and Hogyun Seo contributed equally to this work. Correspondence and requests for materials should be addressed to S.Y.L. (email: leesy@kaist.ac.kr) or to K.-J.K. (email: kkim@knu.ac.kr)

Plastics are essential materials in our lives due to their desirable properties, such as lightness, durability, low price, easy processibility into many different forms, and non-degradability. However, non-degradability, which had been considered to be a great advantage of employing plastics, has been reconsidered as a major cause of environmental problems, in particular due to the accumulation of waste plastics in landfill and ocean. Plastics production has continuously increased and about 320 million tons of plastics were produced globally in 2015[1]. Because most plastics are resistant to biodegradation and require a long time to degrade, the amount of plastic wastes to be accumulated is expected to reach 33 billion tons by 2050[2]. Therefore, much effort has been exerted to reduce plastic wastes. To remove plastic wastes and recycle plastic-based materials, several chemical degradation methods such as glycolysis, methanolysis, hydrolysis, aminolysis and ammonolysis have been developed[3]. However, these methods generally require high temperature and often generate additional environmental pollutants[4]. Alternatively, biocatalytic degradation might be applied as an ecofriendly method. Microbes can degrade plastics with ester bond via enzymatic hydrolysis through colonization onto the surfaces of materials. The degree of biodegradability of plastics depends on their chemical and physical properties[5].

Poly(ethylene terephthalate) (PET) is an extensively and widely used polyester and is also resistant to biodegradation. According to a report by the US National Park Service, PET bottles require approximately 450 years to be decomposed[6]. PET comprises terephthalate (TPA) and ethylene glycol (EG), which are polymerized through ester linkage. Various bacterial hydrolases, such as cutinases[7], lipases, carboxylesterases, and esterases[8], have been shown to degrade PET, although to different extents[9]. Among the PET-degrading enzymes identified to date, *Tf*H and *Tf*H BTA-2 from *Thermobifida fusca* DSM43793, *Tf*Cut1 and *Tf*Cut2 from *T. fusca* KW3, LC cutinase from the metagenome in plant compost, cutinase from *Saccharomonospora viridis* AHK190, *Hi*C from *Thermomyces insolens*, and lipase B from *Candida antarctica* have been shown to possess relatively higher degradability. However, the degradation activities are still too low for industrial applications[9, 10].

To enhance enzymatic activity, several strategies have been adopted. Through site-directed mutagenesis of the active site, cutinases exhibit higher hydrolysis activity[11, 12]. Moreover, the introduction of $Ca^{2+}$ or $Mg^{2+}$ ions to esterases[13] or the addition of disulfide bonds to esterases[14] improves the thermal stability of the enzymes, leading to enhanced PET degradability. Recently, a dual enzyme system consisting of *Tf*cut2 from *T. fusca* KW3 and LC cutinase[15] or lipase from *C. antarctica* and cutinase from *Humicola insolens*[16] was found to have synergistic effects. Despite these attempts, the PET degradation activity still remains low.

Recently, a new bacterial species, *Ideonella sakaiensis*, which can use PET as a carbon source, was isolated[17–20]. The PETase of *I. sakaiensis* (*Is*PETase) can degrade PET at a moderate temperature (30 °C) and has relatively higher activity than other PET-degrading enzymes, such as cutinases and lipases[17]. In addition, *Is*PETase showed higher specificity for PET. The superior capability of *Is*PETase for PET degradation has been receiving much attention. However, the detailed enzyme mechanism has not been elucidated, hampering further studies. Here we report the crystal structure and key structural features of *Is*PETase. While we determined the crystal structure (Protein Data Bank accession code, 5XJH) and this manuscript was under revision, another group independently determined the crystal structure of *Is*PETase (Protein Data Bank accession code, 5XG0)[21]. In this study, based on structural and biochemical studies of *Is*PETase, we propose the detailed molecular mechanism of *Is*PETase, so far the most efficient and more specific PET degrading enzyme, compared with

other PET-degrading enzymes. In addition, we constructed the *Is*PETase variant with enhanced PET-degrading activity by structural-based protein engineering.

## Results

**Overall structure of *Is*PETase.** For structural determination of *Is*PETase, the signal peptide sequences (Met1-Ala33) were removed for the production of the core domain of the protein. The recombinant *Is*PETase protein had additional amino-acid residues at both N and C-termini (Met13–Met33 and Leu291–Gln312) due to the use of the pET15b vector. To elucidate the molecular mechanisms of *Is*PETase, its crystal structure was determined at 1.5 Å resolution (Fig. 1 and Supplementary Fig. 1). The structure reported here comprises residues Ser31-Gln292 visible in the electron density map. The refined structure was in good agreement with the X-ray crystallographic statistics for bond angles, bond lengths, and other geometric parameters (Supplementary Table 1). The asymmetric unit in the $P2_12_12_1$ space group contains one molecule of *Is*PETase, indicating that *Is*PETase exists as a monomer. The size-exclusion chromatography experiment also confirms that *Is*PETase functions as a monomer (Supplementary Fig. 2). The *Is*PETase structure belongs to the α/β hydrolase superfamily[22–24], and the central twisted β-sheet is formed by nine mixed β-strands (β1–β9) and surrounded by seven α-helices (α1–α7; Fig. 1b). As observed in other α/β hydrolase superfamily proteins such as lipases and esterases, *Is*PETase contained the conserved serine hydrolase Gly–x1–Ser–x2–Gly motif (Gly158–Trp159–Ser160–Met161–Gly162) located at the active site (Fig. 1a).

**Active site of *Is*PETase.** *Is*PETase has been shown to degrade PET into monomers such as bis(2-hydroxyethyl) terephthalate (BHET), mono(2-hydroxyethyl) terephthalate (MHET), and TPA[17] (Supplementary Fig. 3). *Is*PETase also hydrolyzes BHET, which is a commercial monomer having similarity with the core structure of PET and has been widely used for studying PET. BHET is hydrolyzed by *Is*PETase to MHET with no further decomposition[17]. In order to elucidate the substrate binding mode of *Is*PETase, we first attempted to determine its structure in complex with BHET. However, neither co-crystallization nor soaking of BHET into the *Is*PETase crystal was successful, potentially because we could not use high concentration of BHET in co-crystallization and soaking due to its low solubility. Alternatively, we speculated the substrate binding mode of the enzyme by covalent docking calculation using 2-hydroxyethyl-(mono-hydroxyethyl terephthalate)$_4$, 2-HE(MHET)$_4$, a four-MHET molecule mimicking PET (Supplementary Fig. 3). At the active site of *Is*PETase, three residues Ser160, His237, and Asp206 form a catalytic triad and Ser160 is postulated to function as a covalent nucleophile to the carbonyl carbon atom in the scissile ester bond, as in other carboxylesterases (Fig. 2a). Oxyanion of the tetrahedral intermediate is stabilized by an oxyanion hole that consists of nitrogen atoms of Tyr87 and Met160 with distances of 2.90 and 2.83 Å, respectively (Fig. 2a). The substrate binding site is simulated to form a long, shallow L-shaped cleft on a flat surface with dimensions of approximately 25 and 29 Å (Fig. 2b–d). The surface of the substrate binding cleft is mainly hydrophobic and the length of the cleft is ~40 Å (Fig. 2b). Based on the scissile ester bond of 2-HE(MHET)$_4$, the substrate binding site can be divided into two subsites, subsite I and subsite II, where one and three MHET moieties are bound, respectively (Fig. 2b,e). For binding of the first MHET moiety in the subsite I, the benzene ring is positioned on a ravine between the two aromatic residues of Tyr87 and Trp185 (Fig. 2b, e). The π–π interactions between Trp185 and the benzene ring of the first MHET moiety with a

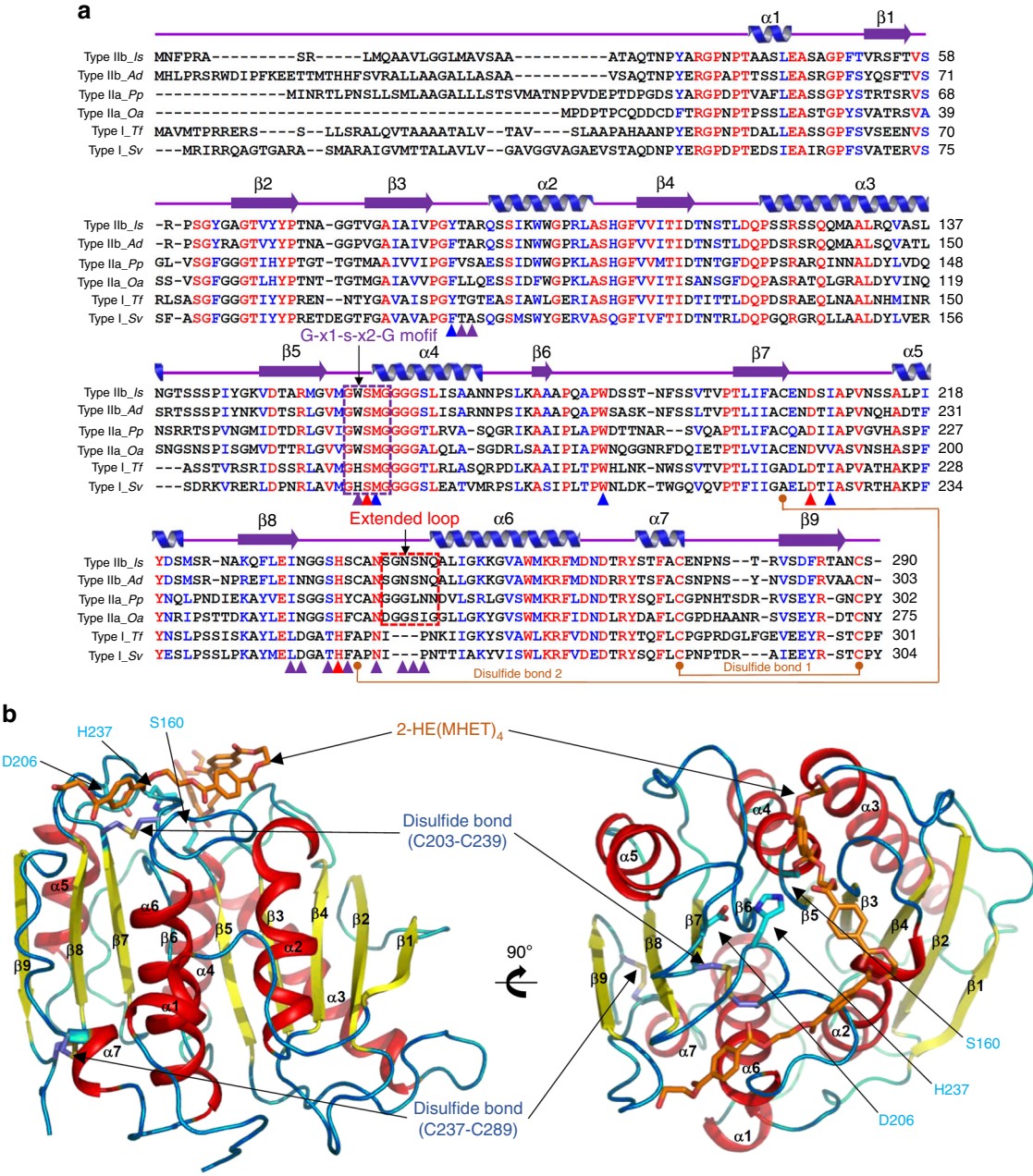

**Fig. 1** Crystal structure of *Is*PETase. **a** Amino-acid sequence alignment of PET-degrading enzymes. Amino-acid sequences of six PET-degrading enzymes, two each of type I, type IIa, and type IIb, are compared. Secondary structure elements are drawn on the basis of the *Is*PETase structure and shown with a purple-colored arrow (β-sheet) and blue-colored helix (α-helix). The Gly-x1-Ser-x2-Gly motif and the extended loop are highlighted as purple and red color boxes, respectively. Residues involved in enzyme catalysis and constitution of subsite I and subsite II are indicated by red-, blue- and purple-colored triangles, respectively. The disulfide bond found in all six enzymes is indicated with an orange-colored line and labeled with 'Disulfide bond 1'. The additional disulfide bond found only in *Is*PETase is also indicated with an orange-colored line and labeled with 'Disulfide bond 2'. *Is*, *Ad*, *Pp*, *Oa*, *Tf*, and *Sv* are representations of PET-degrading enzymes from *Ideonella sakaiensis*, *Acidovorax delafieldii*, *Pseudomonas pseudoalcaligenes*, *Oleispira antarctica*, *Thermobifida fusca*, and *Saccharomonospora viridis*, respectively. **b** Structure of *Is*PETase. The monomeric structure is shown as a ribbon diagram. The three residues of Ser160, Asp206 and His237 forming a catalytic triad are shown as cyan-colored sticks, and the two disulfide bonds are as light-blue-colored sticks. The simulated 2-HE(MHET)$_4$ molecule at the active site is shown as an orange-colored stick. The right side figure is rotated 90° horizontally from the left side figure

distance of ~3.6 Å seem to be a main contributor to the stabilization of the ligand (Fig. 2e). Met161 and Ile208 are also predicted to assist the binding of the first MHET by providing a hydrophobic surface at the bottom and the side of subsite I, respectively (Fig. 2e). Subsite II tends to form a longer and shallower cleft than subsite I and accommodates three MHET moieties (the second, third and fourth MHET moieties of 2-HE

(MHET)$_4$) (Fig. 2b–d). Based on the binding of MHET, the subsite II is further divided into three parts, subsite IIa, IIb, and IIc (Fig. 2b, e). Subsite II is composed of residues including Thr88, Ala89, Trp159, Ile232, Asn233, Ser236, Ser238, Asn241, Asn244, Ser245, Asn246, and Arg280. Although the interaction between subsite II and three MHET moieties seems to be mainly mediated through hydrophobic interactions, carbonyl oxygen

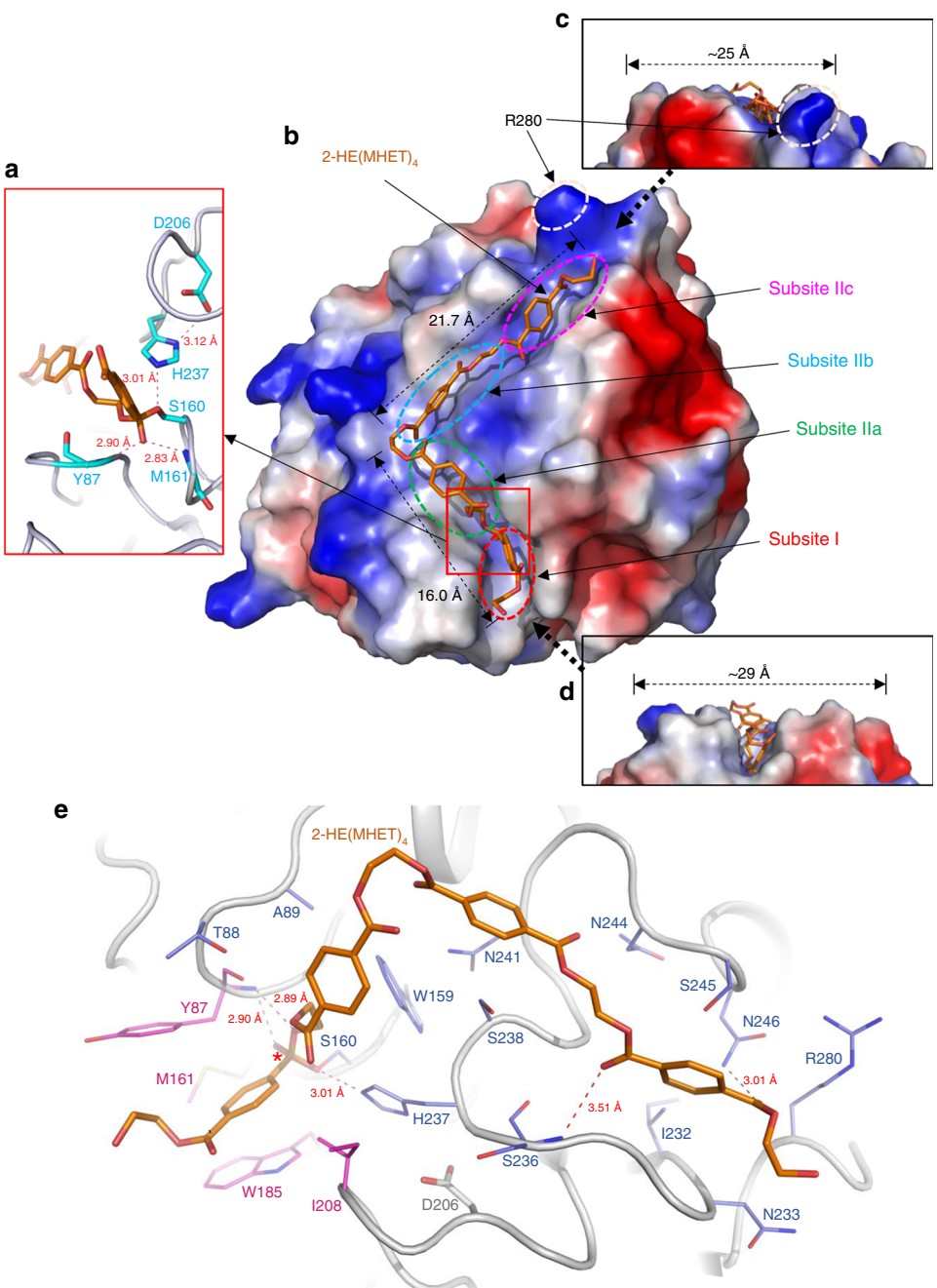

**Fig. 2** Active site of *Is*PETase. **a** Catalytic triad and the docking model of the reaction intermediate of 2-HE(MHET)$_4$ in *Is*PETase. The three residues of Ser160, Asp206 and His237 forming a catalytic triad are shown as cyan-colored sticks and labeled appropriately. The distances of the interaction involved in the oxyanion hole and the catalytic triad are also shown. **b** Substrate binding site of *Is*PETase. The *Is*PETase structure is presented as an electrostatic potential surface model. The 2-HE(MHET)$_4$ docking model is shown as an orange-colored stick, and the cleavage site is highlighted as a red box. Subsite I, IIa, IIb, and IIc of the substrate binding site are indicated with red, green, cyan, and magenta-colored dotted circles, respectively. Arg280 residue at the end of subsite IIc is also indicated. **c**, **d** Side views of the substrate binding mode of *Is*PETase in **b**. **e** Residues involved in the active site of *Is*PETase. The *Is*PETase structure is presented as a cartoon diagram with a gray color. Residues involved in binding of 2-HE(MHET)$_4$ are shown as a line model, and those constituting subsite I and subsite II are distinguished with colors of magenta and light-blue, respectively, and the ester bond that is cleaved by the enzyme is indicated with a star mark. The 2-HE(MHET)$_4$ docking model is shown as an orange-colored stick. The hydrogen bonds formed between the residues and the substrate are shown as red-colored lines

atoms in the fourth MHET moiety also form polar interactions with a main chain of Ser236 and a side chain of Asn246 in the subsite IIc (Fig. 2b, e). Arg280 is located at the end of subsite IIc and the residue seems to hinder the extension of the substrate binding site due to its positive charge and slightly protruding

structure (Fig. 2b, c, e); this residue was further examined by protein engineering (see below).

In order to confirm the residues involved in enzymatic catalysis and substrate binding, site-directed mutagenesis experiments were conducted. First, three catalytic residues, Ser160, Asp206

and His237, were replaced with Ala, and hydrolytic activities were measured using BHET as a substrate. All three variants, $Is$PETase$^{S160A}$, $Is$PETase$^{D206A}$, and $Is$PETase$^{H237A}$, showed almost complete loss of the activity (Fig. 3a), indicating that these three residues were involved in catalysis. Next, Tyr87, Trp185, Met161, and Ile208 residues, which constitute subsite I, were replaced with Ala. Two mutants, $Is$PETase$^{Y87A}$ and $Is$PETase$^{W185A}$, showed only 5% BHET hydrolytic activity compared with $Is$PETase$^{W/T}$ (Fig. 3a). This result indicates that abolishment of the π–π interactions of these residues with the benzene ring of the first MHET moiety severely decreases stabilization of the first MHET moiety. The $Is$PETase$^{M161A}$ and $Is$PETase$^{I208A}$ variants exhibited 52% and 46% activity, respectively, compared with $Is$PETase$^{W/T}$ (Fig. 3a). It indicates that these residues also contribute to the constitution of subsite I, although these residues are not as crucial as Tyr87 and Trp185 residues. We also replaced residues of Trp159, Ser238, and Asn241, which constituted subsite II, with Ala. $Is$PETase$^{W159A}$, and $Is$PETase$^{N241A}$ showed only 8% and 18% BHET hydrolytic activities compared with $Is$PETase$^{W/T}$ (Fig. 3a), suggesting that these residues are crucial in the constitution of subsite II. However, $Is$PETase$^{S238A}$ showed a similar level of BHET hydrolytic activity compared with $Is$PETase$^{W/T}$ (Fig. 3a). The result suggests that replacing of Ser238 with Ala does not seem to affect BHET hydrolytic activity of the enzyme. Next, the PETase activities of the above variants were measured using PET film as a substrate. Variants of the catalytic triad with Ala showed almost complete loss of enzyme activities, and those involved in constitution of the substrate binding site exhibited decreased PETase activities compared with $Is$PETase$^{W/T}$ (Fig. 3b).

As we described above, Arg280 at the end of subsite IIc is a polar residue and shows protruding shape, which seems to hinder stable binding of PET substrate beyond the fourth moiety (Fig. 2b, c, e). Based on these results, we predicted that the substitution of Arg280 into a small hydrophobic residue might allow more stable binding of longer substrate, subsequently leading to an increase in PETase activity. We replaced Arg280 with Ala, and measured both the BHET hydrolytic and the PETase activities. $Is$PETase$^{R280A}$ showed similar activity on BHET hydrolysis compared with $Is$PETase$^{W/T}$ (Fig. 3a), and the result can be explained by the fact that Arg280 is located distal from the catalytic site and thus does not directly participate in substrate binding when BHET is used as a substrate. As hypothesized, $Is$PETase$^{R280A}$ showed increased PETase activity by 22.4% in 18 h and 32.4% in 36 h, compared with $Is$PETase$^{W/T}$, when PET film was used as a substrate (Fig. 3b). To investigate whether the replacement of Arg280 with Ala indeed changed the conformation of substrate binding site (subsite IIc) allowing longer substrate binding, we determined the structure of $Is$PETase$^{R280A}$ at a 1.36 Å resolution (Supplementary Table 1). As expected, compared with $Is$PETase$^{W/T}$, $Is$PETase$^{R280A}$ showed an extended subsite IIc by providing hydrophobic and non-protruding cleft (Fig. 3c). It is interesting that the replacement of Arg280, located distal from the catalytic site with a distance of 22.8 Å, with Ala enhanced the enzymatic activity. This result could not be obtained without reliable docking calculation, which identified unique substrate binding characteristics of $Is$PETase.

**PET degradation mechanism by $Is$PETase.** Based on the structural observations and biochemical studies described above, we propose the following PET degradation process. To start PET degradation, the PETase secreted from the bacterium would first bind to the PET surface using its flat hydrophobic surface that has a substrate binding cleft (Fig. 2b–d). The PET degradation process can be divided into two steps, nick generation step and

terminal digestion step. In the nick generation step, four MHET moieties are bound to each subsite (one MHET moiety to subsite I and three MHET moieties to subsite II) and the scissile ester bond seems to be positioned between subsite I and II near the catalytic Ser160 residue (Fig. 4a). Then, the cleavage of one ester bond causes the formation of a nick in PET, resulting in generation of two PET chains with different terminals: TPA-terminal released from subsite I and HE-terminal released from subsite II (Fig. 4a).

In the terminal digestion step, two PET chains having the HE- and the TPA-termini are digested into MHET monomers in somewhat different manners. For digestion of PET having the HE-terminal ($^{HE}$PET), the terminal MHET and the next three MHET moieties bind to subsite I and subsite II, respectively, and breakage of the ester bond results in the production of one MHET monomer and $^{HE}$PET$_{n-1}$ (Fig. 4b). Subsequent digestion of $^{HE}$PET$_{n-1}$ is expected to occur in a manner similar to that of the first cleavage process. Digestion of PET having the TPA-terminal ($^{TPA}$PET) is also expected to occur through positioning of the terminal TPA and the next three MHET moieties at subsite I and subsite II, respectively (Fig. 4b). Cleavage of the ester bond seems to produce one TPA molecule and $^{HE}$PET$_{n-1}$, and this $^{HE}$PET$_{n-1}$ undergoes subsequent cleavage as observed in the $^{HE}$PET degradation process (Fig. 4b). Alternatively, $^{HE}$PET and $^{TPA}$PET can also be digested though binding of PET polymer chains and the enzyme in the reverse direction, although this type of digestion might be less efficient than the above digestion. In this case, one or two MHET moieties, instead of three MHET moieties, can bind to subsite II (Fig. 4b). These bindings can produce a variety of PET monomers and dimers such as 2-HE (MHET)$_2$, (MHET)$_2$, MHET and BHET, which can be finally digested to MHET, TPA and EG (Fig. 4b). Continuous digestions of $^{HE}$PET and $^{TPA}$PET proceed in a combinatorial manner, as described above, resulting in accumulation of four molecules, including MHET, TPA, BHET, and EG (Fig. 4b). BHET can be further degraded into MHET and EG, and finally, three molecules, MHET, TPA, and EG, accumulate due to PET degradation (Fig. 4b). In addition, it is worth to note that degradation of PET film by $Is$PETase accumulates significant amount of TPA (Fig. 2b), although $Is$PETase can not hydrolyze MHET to TPA and EG[17]. Based on the PET degradation process we propose here, it can be also concluded that accumulation of TPA from PET film degradation is mainly derived from terminal digestion step of $^{TPA}$PET.

**Structural comparison with other PET degrading enzymes.** Structural comparison using the DALI server[25] showed that the structure of $Is$PETase is quite similar to those of cutinases from *T. fusca* KW3 (*Tf*Cut2, PDB code 4CG1, Z-score 42.4), *S. viridis* (*Sv*Cut, PDB code 4WFJ, Z-score 42.3), and *Thermobifida alba* (*Ta*Cut, PDB code 3VIS, Z-score 42.1). These structural homologs have been identified to have a PET-degrading activity[26–28] and share ~50% amino-acid identity with $Is$PETase. In order to provide a structural basis for why $Is$PETase shows much higher PETase activity than these other PET-degrading enzymes, we compared the structure of $Is$PETase with other three PET-degrading enzymes. As residues constituting the substrate binding site are almost conserved within these enzymes, the structure of $Is$PETase was compared with that of *Tf*Cut2, a representative cutinase studied for PET degradation.

Three residues constituting the Ser–His–Asp catalytic triad are located at the same positions in *Tf*Cut2 (Fig. 5a), indicating that these enzymes catalyze PET degradation through the same catalytic mechanism. The residues constituting subsite I are also identical in both $Is$PETase and *Tf*Cut2. It suggests that the

binding mode of the first MHET moiety to subsite I is similar in both enzymes (Fig. 5a). However, significant structural differences were observed in the conformation of subsite II. In *Tf*Cut2, His169 and Phe249 residues are located at the corresponding

positions of Trp159 and Ser238 in subsite II of *Is*PETase, respectively (Fig. 5a). To verify whether the residues, Trp159 and Ser238, play a crucial role in the high PET-degrading activity of *Is*PETase, these two residues were replaced with His and Phe,

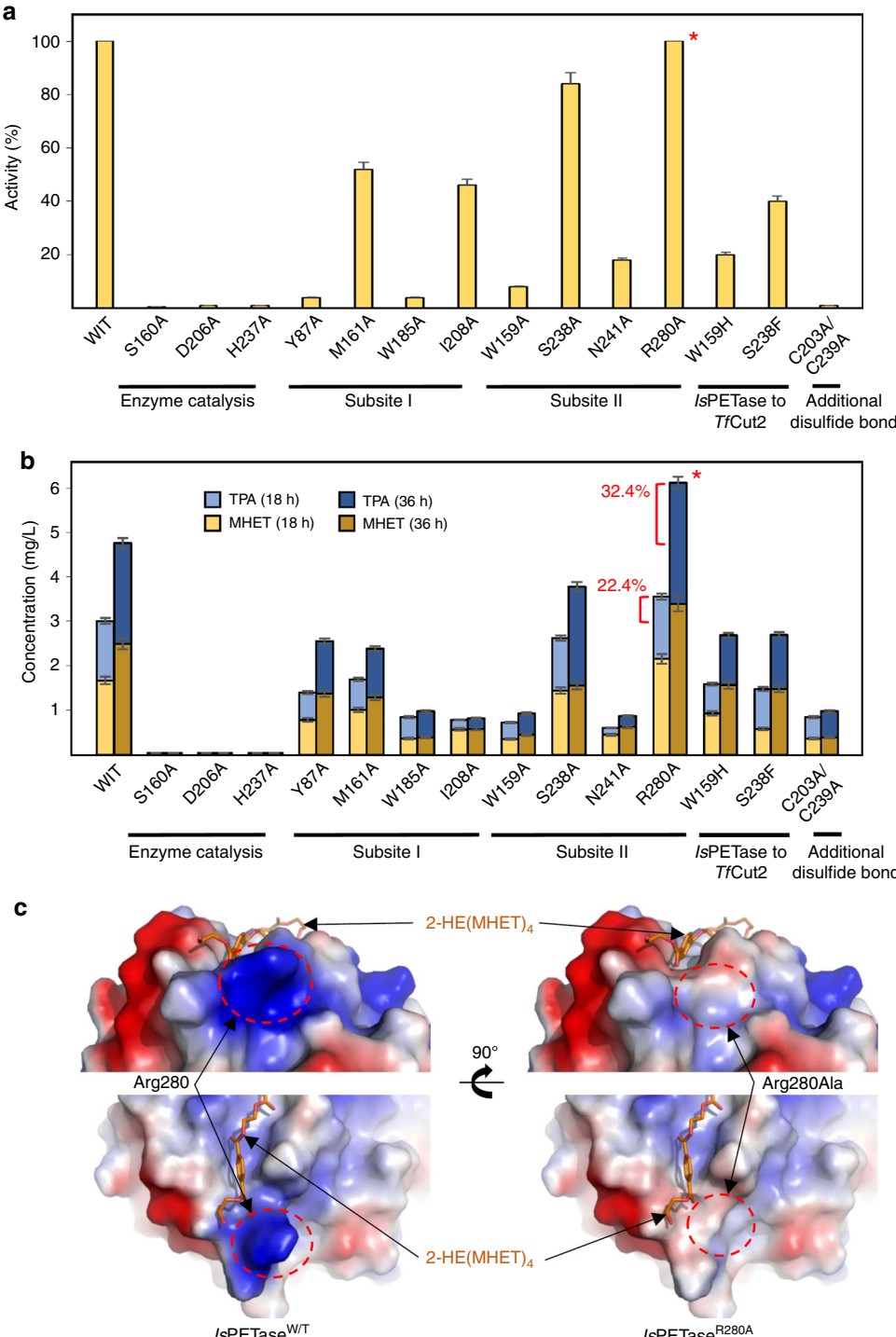

**Fig. 3** PETase activity of the variants. **a** Hydrolytic activities of *Is*PETase and its variants using BHET as a substrate. PETase activities of *Is*PETase and its variants were measured using BHET concentration of 200 mg L$^{-1}$ and enzyme concentration of 50 nM. The amount of produced MHET was monitored by HPLC analysis. The PETase activities of the *Is*PETase variants were compared with that of the wild-type. **b** PETase activity of *Is*PETase proteins using the PET film as a substrate. PET film degradation activity of *Is*PETase proteins were measured using enzyme concentration of 200 nM. The amount of produced MHET and TPA was monitored by HPLC analysis. The PETase activities of the *Is*PETase variants were compared with that of the wild-type. The *Is*PETase$^{R280A}$ variant showing an increased activity is highlighted with a star mark. **c** Electrostatic potential surface presentation of *Is*PETase$^{R280A}$ structure. The 2-HE(MHET)$_4$ molecule is labeled. The Arg280 residue in *Is*PETase$^{W/T}$ and the Arg280Ala (R280A) change in *Is*PETase$^{R280A}$ are indicated with dotted circles. Error bars represent the s.d. values obtained in duplicate experiments

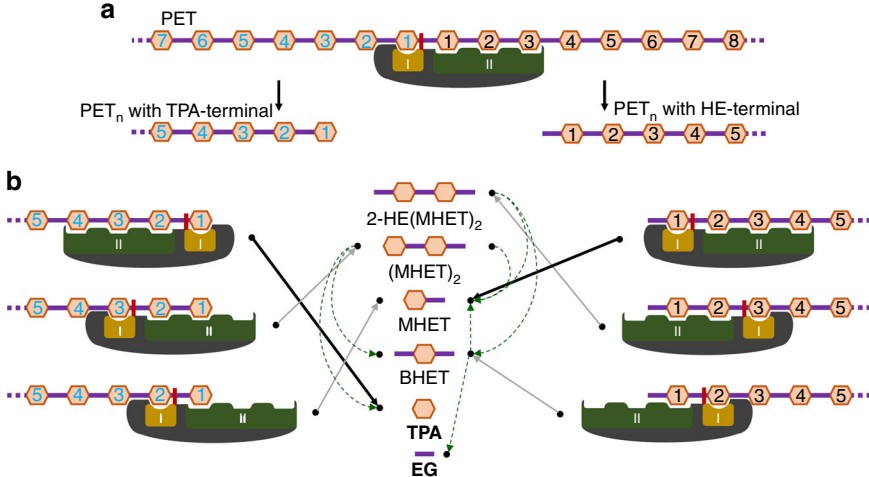

**Fig. 4** Schematic diagram of PET degradation process. **a** Nick generation step. The TPA and EG moieties of the PET polymer are presented with orange-colored hexagons and purple-colored lines, respectively. *Is*PETase is shown with a dark-gray color diagram. Subsite I and subsite II of *Is*PETase are shown as an orange and green-colored diagrams, and labeled as I and II, respectively. The catalytic Ser160 residue is shown as a red-colored rectangle. The PET$_n$ polymers with the TPA-terminal and with the HE-terminal are labeled. The TPA moieties of these PET$_n$ polymers are labeled by numbers 1, 2, 3, 4, and 5 from each terminal. **b** Terminal digestion step. Each enzymatic reactions in terminal digestion step is shown as a diagram. The cleaved products from the enzymatic reactions are indicated with black-colored arrows. The light green-colored dotted lines indicate the paths the PET polymer products go for the next reaction. Six PET-related compounds, 2-HE(MHET)$_2$, (MHET)$_2$, MHET, TPA, BHET, and EG, that are cleaved products from terminal digestion step, are shown. The final degradation product (MHET, TPA and EG) are labeled in bold

respectively, the corresponding residues in *Tf*Cut2. The *Is*PETase$^{W159H}$ and the *Is*PETase$^{S238F}$ variants showed dramatically decreased hydrolytic activities from both uses of BHET and PET as a substrate (Fig. 3a, b). The differences in these residues seem to make subsite IIa of *Tf*Cut2 narrower and deeper than that of *Is*PETase, resulting in reduced accessibility of the second MHET moiety to its binding site in *Tf*Cut2 (Fig. 5b, c). Furthermore, more striking structural difference was observed in the connecting loop of β8-α6 (Figs. 1a and 5a). Compared with *Tf*Cut2, *Is*PETase has an extended loop in the region with three extra-residues (Asn244, Ser245, and Asn246; Figs. 1a and 5a). Interestingly, the unique conformation of the extended loop in *Is*PETase allows the formation of subsite IIb and IIc by constituting a continuous cleft from subsite IIa (Fig. 5b, c). On the other hand, the conformation of the region in *Tf*Cut2 prevents the formation of subsite IIb and IIc by blocking the constitution of the continuous cleft (Fig. 5b,c).

In addition to structural differences in subsite II, existence of two disulfide bonds is another important structural feature of *Is*PETase. In other PET-degrading enzymes, one disulfide bond is observed near the C-terminal; the disulfide bond is formed between Cys281 and Cys299 in *Tf*Cut2, between Cys287 and Cys302 in *Sv*Cut, and between Cys276 and Cys294 in *Ta*Cut (Fig. 5d). The disulfide bond is also conserved in *Is*PETase and formed between Cys273 and Cys289 (Fig. 5d). Since the disulfide bond is located at the opposite side of the active site, it can be assumed that the disulfide bond has no direct effect on the enzyme activity, but rather influences the structural stability of the enzyme. Interestingly, *Is*PETase has an additional disulfide bond between Cys203 and Cys239 in the vicinity of the active site (Fig. 5e). However, all other PET-degrading enzymes have Ala residues at the corresponding positions (Fig. 5f). Since internal disulfide bonds tend to increase the thermal stability of proteins, another *Is*PETase variant without the additional disulfide bond was generated to investigate how the disulfide bond affects the thermal stability of *Is*PETase. The Tm values of *Is*PETase$^{W/T}$ and *Is*PETase$^{C203A/C239A}$ variant were 46.8 and 33.6 °C, respectively (Supplementary Fig. 4), suggesting that the additional disulfide

bond plays an important role in the thermal stability of *Is*PETase. As expected, the PETase activity of *Is*PETase$^{C203A/C239A}$ was dramatically decreased compared with that of *Is*PETase$^{W/T}$ (Fig. 3a, b). The Tm of *Tf*Cut2 was measured to be 67.9 °C, which is much higher than that of *Is*PETase$^{W/T}$. The reason with high Tm value of *Tf*Cut2 is expected as *T. fusca* is a thermophilic bacterium, and its high thermal stability is due to other structural features of the protein even without the additional disulfide bond.

**Phylogenetic tree analysis.** Having understood the reasons for the much higher PETase activity of *Is*PETase compared with other known PET-degrading enzymes as described above, we became interested in comparatively analyzing all possible 69 PETase-like enzymes from phylogenetically diverse organisms. For this, a maximum-likelihood phylogenetic tree was constructed (Fig. 6a and Supplementary Fig. 5). PETase-like enzymes can be classified into two types, type I and type II. Fifty-seven enzymes, including *Tf*Cut2, belong to type I, and the remaining twelve enzymes including *Is*PETase belong to type II. Type II PET-degrading enzymes can be further classified into two subtypes, type IIa and type IIb. Among 12 type II enzymes, eight enzymes belong to type IIa and four enzymes including *Is*PETase belong to type IIb. In all 69 proteins, three residues constituting the catalytic triad, such as Ser, His, and Asp, are conserved (Fig. 6b), indicating that these enzymes have the same catalytic mechanism. The residues constituting subsite I are also highly conserved in all proteins, suggesting that the binding mode of the first MHET moiety to subsite I is quite similar among these proteins (Fig. 6b). However, there are major differences in key residues comprising subsite II and in the presence of additional disulfide bond depending on the type of enzyme. In type I PET-degrading enzymes, there is no additional disulfide bond and the extended loop found in *Is*PETase. Moreover, all of these enzymes possess His and Phe/Tyr residues at the corresponding positions of Trp159 and Ser238 in *Is*PETase, respectively (Fig. 6b). These structural features suggest that type I PET-degrading enzymes have much lower PET-degrading activity compared with

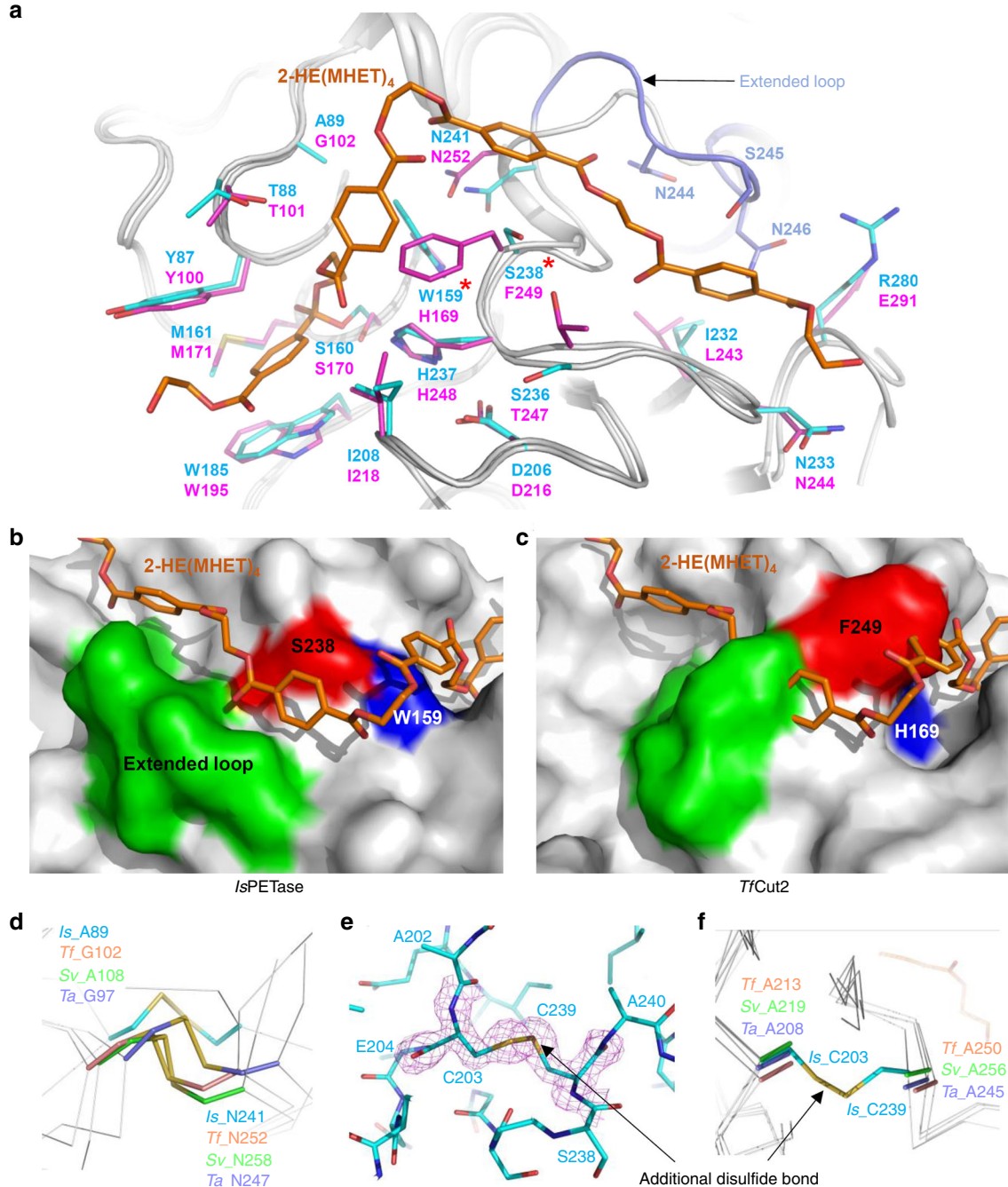

**Fig. 5** Structural comparison of PET-degrading enzymes. **a** Superposition of the structures of *Is*PETase and *Tf*Cut2. Both structures of *Is*PETase and *Tf*Cut2 are shown as gray-colored cartoon model. The 2-HE(MHET)$_4$ docking model of *Is*PETase is shown as an orange-colored stick. Residues constituting subsite I and subsite II of *Is*PETase and *Tf*Cut2 are shown as a line models with cyan and magenta colors, respectively, and labeled. The unique residues observed in *Is*PETase, Trp159 and Ser238, are indicated with star marks. The extended loop of *Is*PETase distinguishable from *Tf*Cut2 is shown in light-blue color. **b, c** Structural difference on subsite II of *Is*PETase (**b**) and *Tf*Cut2 (**c**). The structures of *Is*PETase and *Tf*Cut2 enzymes are presented as surface models with a gray color. The 2-HE(MHET)$_4$ docking model of *Is*PETase is shown as an orange-colored stick. The Ser328 and Trp159 residues in subsite II and extended loop of *Is*PETase corresponding phenylalanine and histidine residues and extended loop in other PET-degrading enzymes are distinguished and labeled, respectively. **d** Disulfide bond found in PET-degrading enzymes. The disulfide bond found in each all four PET-degrading enzymes is shown as a stick model, respectively and the residues forming the disulfide bond are labeled. **e, f** Additional disulfide bond found in *Is*PETase. The *Is*PETase structure is presented as a stick model and the omit electron densities (magenta mesh) of the residues constituting the additional disulfide bond in *Is*PETase are contoured at 2.0 $\sigma$ (**e**). The additional disulfide bond region in *Is*PETase is compared with the corresponding regions in other PET-degrading enzymes (**f**). The residues forming the additional disulfide bond in *Is*PETase and those located at the corresponding positions in other PET-degrading enzymes are shown as a stick model and labeled appropriately

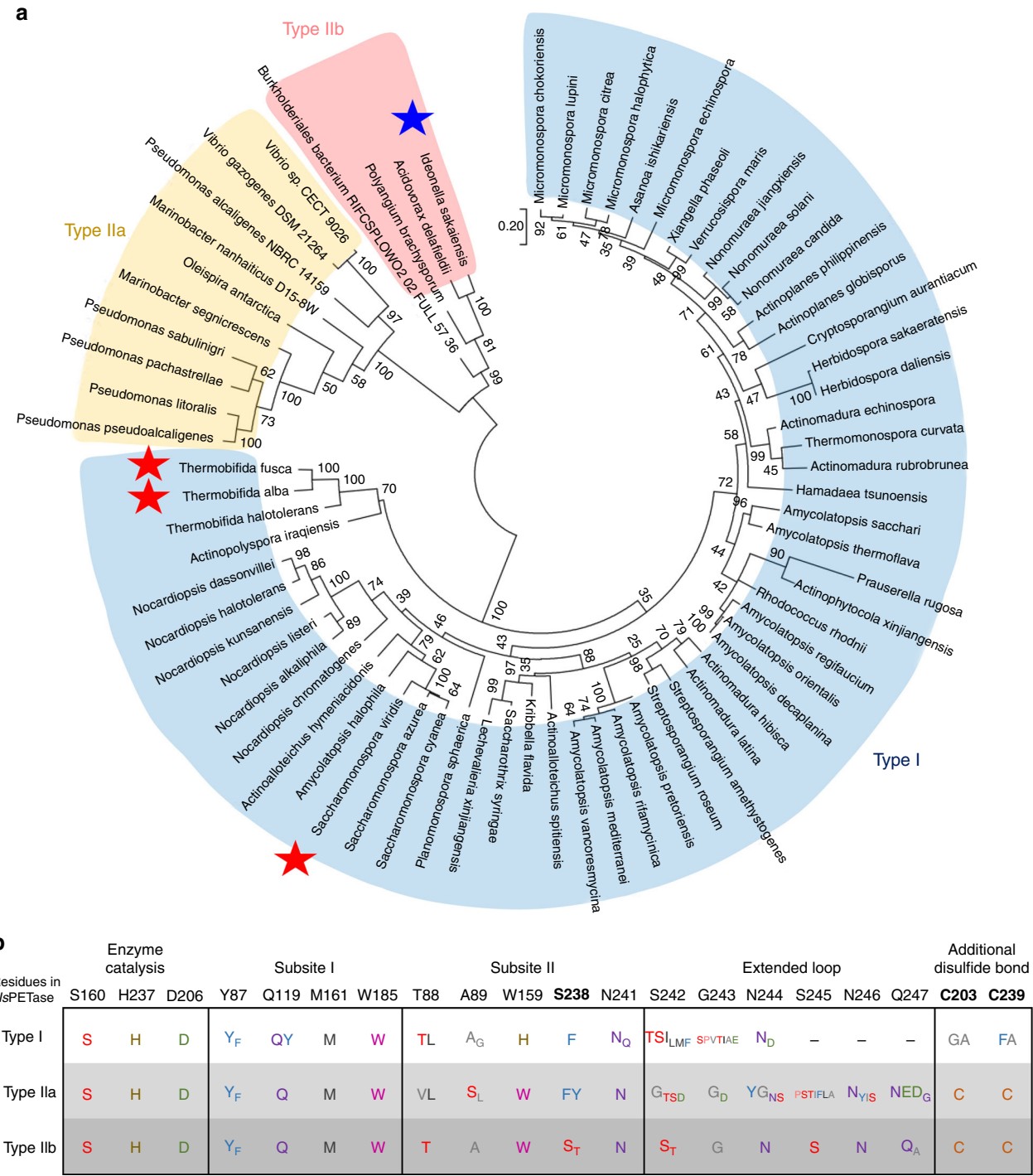

**Fig. 6** Phylogenetic tree and comparative analysis of PET-degrading enzymes. **a** Unrooted maximum likelihood tree of the PET-degrading enzymes. The phylogenetic tree was drawn as a circle model. Bootstrap values are shown at each node as percentage of 100 replicates. Type I, type IIa, and type IIb of the PET-degrading enzymes are labeled as different color schemes. *Is*PETase is indicated by a blue-colored star, and other three PET-degrading enzymes whose structures are determined are indicated by a red-colored star. **b** Amino-acid sequence alignment of key residues in PET-degrading enzymes. The key residues involved in the enzyme catalysis, the constitution of subsite I and subsite II, the extended loop, and the additional disulfide bond in *Is*PETase aligned

*Is*PETase. Unlike type I enzymes, all type II PET-degrading enzymes have additional disulfide bonds and the extended loop (Fig. 6b). However, substantial differences were observed in residues constituting subsite II and the extended loop depending on type IIa and type IIb (Fig. 6b). Although residues constituting subsite II and the extended loop are conserved among type IIb enzymes, type IIa enzymes have a Phe or Tyr residue at the position of Ser238 in *Is*PETase (Fig. 6b). Because the *Is*PETase<sup>S238F</sup> variant exhibited much lower PET-degrading activity compared with *Is*PETase<sup>W/T</sup>, the type IIa proteins are predicted to have lower PET-degrading activities compared with *Is*PETase. Furthermore, type IIa enzymes have highly variable residues at the extended loop (Fig. 6b), implying that environment around subsite IIb and IIc of type IIa enzymes might be quite different

from that of type IIb enzymes. Based on the same reasoning, the other enzymes of type IIb are predicted to have PET-degrading activities similar to that of *Is*PETase. In addition to *Is*PETase, enzymes originating from bacteria, such as *Acidovorax delafieldii*[29], *[Polyangium] brachysporum* DSM 7029[30], and *Burkholderiales* bacterium RIFCSPLOWO2_02_FULL_57_36[31], belong to type IIb. Interestingly, all four bacteria of type IIb enzymes belong to the order *Burkholderiales*, suggesting that these four bacteria seem to have similarly evolved.

## Discussion

Until recently, enzymes using PET as a natural substrate have not been identified, and PET degradation studies have mainly been performed using cutinase and lipase family enzymes. However, degradation of PET with these enzymes was not effective due to their low affinities to PET, which leads low PET degrading activity. Recently, PETase from *I. sakaiensis* was reported to have much higher PET degradation efficiency than those enzymes previously examined[17]. In this work, we determined the crystal structure and reported the structural features conferring high PET-degrading activity on *Is*PETase based on the docking calculations.

While this paper of ours was under revision process, Han et al.[21] also reported the crystal structure of *Is*PETase and its catalytic mechanism. Since they failed to obtain complex structure of *Is*PETase with various ligands, they ended up using inactive variants of *Is*PETse instead of wild-type *Is*PETase and succeeded in making complex structures of *Is*PETase variant (Ser131Ala and Arg103Gly) with two ligand, 1-(2-hydroxyethyl) 4-methyl terephthalate (HEMT) and *p*-nitrophenol (pNP), respectively[21]. The substrate-binding mode in complex with HEMT or pNP in the first TPA binding site, corresponding to the first MHET moiety in this study, agrees with what we described in this study. Based on the complex structure, they focused on the wobbling tryptophan and serine located near the active site. Because we also observed multi-occupancy of Trp156 (Trp185 in our study), this suggested mechanism indicated by the reduced activity of S185H variant is interesting.

On the other hand, we performed docking calculation using a longer substrate, 2-HE(MHET)$_4$ and showed a substrate of four MHET moieties is accommodated by the enzyme. Based on the docking calculation, we thoroughly investigated the substrate binding site (subsite I, IIa, IIb, and IIc) by site-directed mutagenesis and concluded that the superior PET-degrading activity of *Is*PETase is attributed to the differences in the subsite II and disulfide bond formation. In addition, we performed structure-based protein engineering by replacing Arg280 (a residue located quite distal from the catalytic site) to Ala. The Arg280Ala variant showed much higher PET-degrading activity. Then, the crystal structure of this variant (Arg280Ala) was solved as well, which showed that the structure was altered to better accommodate PET substrate as we hypothesized. This is an important finding as the structure-based engineering of a residue (Arg280), which is located far away from the catalytic site with a distance of ~ 23 Å could be selected for enhancing the PETase activity. The proof-of-concept protein engineering demonstrated in this paper based on the 3D structure of *Is*PETase will be invaluable for further rational protein engineering. Moreover, we also provided comparative analysis of all possible 69 PETase-like enzymes from phylogenetically diverse organisms and could suggest PETase candidates potentially having high PET-degrading activities using phylogenetic tree analysis.

Based on this study, we propose future studies for PET degradation in the following two directions. First, it will be necessary to characterize PET degradation by other type IIb enzymes. Second, for actual applications on PET degradation and/or recycling, protein engineering studies toward further enhanced enzyme activity, specificity, and stability are also needed. Also, it is expected that the approaches taken in this study can be extended to studying other enzymes capable of degrading different plastics.

## Methods

**Production of *Is*PETase**. The *Is*PETase gene was amplified by polymerase chain reaction (PCR) using synthesized gene with codon optimization for expression in *Escherichia coli* cells as a template (Supplementary Table 2). The nucleotide sequence corresponding to the signal peptide was removed from the synthetic DNA. The PCR product was then subcloned into pET15b, and the resulting expression vector pET15a: *Is*PETase was transformed into the *E. coli* strain Rosetta gami-B, which was grown in 11 l of lysogeny broth medium containing Ampicillin at 37 °C. After induction by the addition of 1 mM isopropyl β-D-1-thiogalacto-pyranoside, the culture was further incubated for 16 h at 18 °C. The cells were then harvested by centrifugation at 4000 × *g* for 10 min at 20 °C. The cell pellet was resuspended in buffer A (50 mM Na$_2$HPO$_4$-HCl, pH 7.0 and 100 mM NaCl) and then disrupted by ultrasonication. The cell debris was removed by centrifugation at 13,500 × *g* for 20 min, and the supernatant was applied to a Ni-NTA agarose column (Qiagen). After washing with buffer A containing 30 mM imidazole, the bound proteins were eluted with 300 mM imidazole in buffer A. Finally, trace amounts of contaminants were removed by size-exclusion chromatography using a Superdex 200 prep-grade column (320 ml, GE Healthcare) equilibrated with buffer A. All purification steps were performed at 4 °C. The degree of protein purity was confirmed by sodium dodecyl sulfate polyacrylamide gel electrophoresis. The purified protein was concentrated to 28 mg ml$^{-1}$ in 50 mM Na$_2$HPO$_4$-HCl, pH 7.0 and 100 mM NaCl. Site-directed mutagenesis experiments were performed using the Quick Change site-directed mutagenesis kit (Stratagene). The expression and purification of variants of *Is*PETase and *Tf*Cut2 were performed under the same conditions as those used for the native protein of *Is*PETase. To make sure there are no changes in folding of the all-variant, circular dichroism experiment was performed (Supplementary Fig. 7).

**Crystallization of *Is*PETase**. Crystallization of the purified protein was initially performed with the following crystal screening kits: Index and PEG/Ion (Hampton Research) and Wizard I and II (Rigaku) using the hanging-drop vapor-diffusion technique at 20 °C. Each experiment consisted of 1.0 μl of protein solution and 1.0 μl of reservoir solution and then was equilibrated against 50 μl of the reservoir solution. The best quality *Is*PETase crystals appeared in 0.1 M ammonium acetate, 0.1 M bis-tris and 17% Polyethylene glycol 10,000. The crystals were transferred to a cryoprotectant solution containing 0.1 M ammonium acetate, 0.1 M bis-tris (pH 5.0), 20% Polyethylene glycol 10,000 and 30% (v v$^{−1}$) glycerol, extracted with a loop larger than the crystals, and flash-frozen by immersion in liquid nitrogen. Crystallization of *Is*PETase$^{R280A}$ was performed using procedure similar to *Is*PETase$^{W/T}$.

**Data collection and structure determination of *Is*PETase**. Data were collected at 100 K at Beamline 6D at the Pohang Accelerator Laboratory (Pohang, Korea). The data were then indexed, integrated, and scaled using the HKL2000 software suite[32]. The *Is*PETase crystals belonged to the space group P2$_1$2$_1$2$_1$, with unit cell parameters of *a* = 43.48 Å, *b* = 50.40 Å, and *c* = 129.49 Å. With one molecule of *Is*PETase per asymmetric unit, the Matthews coefficient was 2.64 Å$^3$·Da$^{−1}$, which corresponds to a solvent content of 53.38%[33]. The structure of *Is*PETase was determined by molecular replacement with the CCP4 version of MOLREP[34] using the structure of cutinase from *Thermobifida alba* (*Ta*Cut, PDB code 3VIS, 50% sequence identity) as a search model. The model building was performed using the WinCoot program[35] and the refinement was performed with REFMAC5[36]. The data statistics are summarized in Supplementary Table 1. X-ray diffraction data of *Is*PETase$^{R280A}$ crystal were collected at 100 K at Beamline 7A at the Pohang Accelerator Laboratory (Pohang, Korea)[37]. The *Is*PETase$^{R280A}$ crystal also belonged to the space group P2$_1$2$_1$2$_1$, with cell parameters similar to those of *Is*PETase$^{W/T}$ crystal. The structure of *Is*PETase$^{R280A}$ was determined by molecular replacement using the structure of *Is*PETase$^{W/T}$ as a search model. The model building and structure refinement were performed as in *Is*PETase$^{W/T}$. The data statistics are summarized in Supplementary Table 1. The refined models of *Is*PETase and *Is*PETase$^{R280A}$ have been deposited in the Protein Data Bank with PDB code 5XJH and 5YNS, respectively.

**Molecular docking calculations**. Molecular docking of the tetrahedral intermediate from 2-HE(MHET)$_4$ to *Is*PETase structures was carried out by mixed approaches of flexible and covalent docking using AutoDock4.2[38] and AutoDock Vina[39]. The ligand molecule of *Is*PETase was prepared with WinCoot[35] and ProDrg[40] and nonpolar H atoms were merged onto both the ligands and the targets using AutoDockTools prior to performing the docking. For the generation of pdbqt files of both rigid and flexible receptor, flexible residues (Tyr87, Trp159, Ser160, Met161, Trp185, Ile208, His237, Ser238, and Asn241) were selected, and the bonds

in the side chain of each residues were allowed to rotate. The grid box was centered at $x$: −3.249, $y$: 25.239 and $z$: −29.093 with sizes of 90.7, 74.7, and 122.7 Å, respectively. Prior to the covalent docking, non-covalent docking calculation using AutoDock Vina was performed, and nine output poses were generated with their calculated free energy of binding from its own scoring function. The best docking model with the lowest binding energy (−7.1 kcal mol$^{-1}$) was selected, and the conformation of the model was used as an evaluation standard for the following calculation. Furthermore, the induced conformation of the flexible residues in the best model was applied to the receptor for covalent docking. Then, the covalent docking using AutoDock was conducted according to the previous report[41]. A total of 200 docking poses were evaluated based on the proper distances of the oxyanion hole, and the best pose with the binding energy of −10.27 kcal mol$^{-1}$ (from the semi-empirical free energy force field of AutoDock) was selected by similarity to the non-covalent docking result. The docking pose was finally minimized using OPLS3 force field[42] in the Schrödinger suite.

**PETase in vitro assay using bis-hydroxyethyl terephthalate**. To compare the activity of the variants of *Is*PETase, bis-hydroxyethyl terephthalate (BHET) was chosen as a substrate for enzyme assay. The BHET stock solution was prepared by dissolving 2.5 g l$^{-1}$ of BHET in dimethyl sulfoxide. The assay protocols were based on the previously reported paper[17].The enzyme assay was performed in buffer solution (80 mM Na$_2$HPO$_4$ −HCl, 40 mM NaCl) at pH 7.0 with 200 mg l$^{-1}$ of BHET. The enzyme reaction was started by the addition of 50 nM enzyme and was kept at 30 °C for 30 min. Then, the reaction was terminated by heating at 85 °C for 15 min. The reaction mixtures were centrifuged at 13,200 r.p.m. for 10 min. Lastly, the supernatant was applied to LC analysis.

**PETase in vitro assay using PET film**. *Is*PETase assays were performed as previously reported[17] with slight modifications described below. To analyze the degradation rate of PET by PETases, commercial PET film (UBIGEO, Korea) was used as the substrate for enzyme assay. The PET film was prepared in a circular form with 6 mm diameter. The PET film was soaked in 300 μl of pH 9.0 glycine-NaOH buffer with 200 nM of enzyme. The reaction mixture was incubated at 30 °C for 18 and 36 h. The enzyme reaction was terminated by heating at 85 °C for 15 min. Then, the samples were centrifuged at 13,200 r.p.m. for 10 min, and the supernatant was analyzed by LC. After the enzyme reaction, the film was washed with 1% SDS and 20% ethanol in distilled water.

**Analytical methods**. The in vitro assay samples were analyzed by HPLC (1100 Series HPLC, Agilent) equipped with MS (LC/MSD VL, Agilent). Eclipse Plus-C18 column (5 μm, 4.6 × 150 mm, Agilent) was used. All analyses were operated at room temperature (25 °C). For the mobile phase, buffer A (0.1% formic acid in distilled water) and buffer B (acetonitrile) was used at a flow rate of 0.8 ml min$^{-1}$. The mobile phase was changed gradually from 95% buffer A to 30% buffer A at 20 min (all in vol%). The chemicals (BHET, MHET and TPA) were detected at 260 nm.

**Melting temperature (Tm) measurement**. Thermal stability of *Is*PETase$^{W/T}$, *Is*PETase$^{C203A/C239A}$ and *Tf*Cut2 proteins was determined by measuring melting curves at both pH 7.0 and pH 9.0 with the Protein thermal shift dye (Applied Biosystems) in a StepOnePlus Real-Time PCR (Thermo Fisher Scientific) according to manufacturer's instructions. Briefly, 1 μg of protein was mixed with 1× protein thermal shift dye (Applied Biosystems) in 20 μl and signal changes reflecting protein denaturation were monitored by increasing temperature from 25 to 90 °C. Melting temperatures were determined from the first derivative curve.

**Phylogenetic tree analysis**. Iterative searching for PETase-like proteins was performed by Basic Local Alignment Search Tool (BLAST) in National Center for Biotechnology Information (NCBI) server using position-specific iterated BLAST (PSI-BLAST) method. Multiple alignment was performed by Clustal omega. The evolutionary history was inferred by using the Maximum Likelihood method based on the Le_Gascuel_2008 model[43]. The tree with the highest log likelihood (−13243.9235) is shown. The percentage of trees in which the associated taxa clustered together is shown next to the branches. Initial tree(s) for the heuristic search were obtained automatically by applying Neighbor-Join and BioNJ algorithms to a matrix of pairwise distances estimated using a JTT model, and then selecting the topology with superior log likelihood value. The rate variation model allowed for some sites to be evolutionary invariable ([+I], 14.9681% sites). The tree is drawn to scale, with branch lengths measured in the number of substitutions per site. The analysis involved 69 amino-acid sequences. All positions with <95% site coverage were eliminated. That is, fewer than 5% alignment gaps, missing data, and ambiguous bases were allowed at any position. There were a total of 252 positions in the final dataset. Evolutionary analyses were conducted in MEGA7[44].

**Far-UV circular dichroism**. Far-UV (190–260 nm) CD experiments for *Is*PETase and its variants developed this study were carried out in a Jasco J-815 CD Spectrometer (JASCO Corporation, Japan). Scans were recorded at 25 °C between 190 and 360 nm as an average of three scans with 0.5 nm step size, 1.5 s dwell time in a

10 m path length demountable Suprasil quartz cell (Hellma Ltd, UK) and smoothed to obtain the final data of the variants of *Is*PETase (1 mg ml$^{-1}$). Spectra were collected at 1.0-nm intervals with a bandwidth of 1 nm in a buffer containing 10 mm potassium sodium phosphate pH 7.0 in a 1 cm quartz cuvette (Supplementary Fig. 6).

**Data availability**. The refined models of *Is*PETase and *Is*PETase$^{R280A}$ have been deposited in the Protein Data Bank (www.rcsb.org/) with PDB code 5XJH and 5YNS, respectively. Data supporting the findings of this study are available within the article (and its Supplementary information files) and from the corresponding author upon reasonable request.

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

## Acknowledgements
We thank Dr. Oh-Hyun Kwon, the Chairman of Samsung Advanced Institute of Technology for his insight and advice on the need for developing strategies for efficient degradation and recycling of plastics. This work was supported by the Technology Development Program to Solve Climate Changes on Systems Metabolic Engineering for Biorefineries from the Ministry of Science and ICT (MSIT) through the National Research Foundation (NRF) of Korea (NRF-2012M1A2A2026556 and NRF-2012M1A2A2026557). Experiments at PLS-II 6D beamline were supported in part by UCRF, MSIT and POSTECH.

## Author contributions
S.Y.L. and K.-J.K. conceived the project. S.J., I.J.C., H.S., and K.-J.K. designed research. S. J., I.J.C., and H.S. performed research. T.J.S. performed X-ray crystallographic experiment. H.F.S., H.-Y.S., and S.Y.C. analyzed the data. S.J., I.J.C., H.S., and S.Y.L. wrote the paper.
