## [Peer Review File · Nature Communications]

Reviewers' comments:

Reviewer #1 (Remarks to the Author):

In this paper Joo et al. describe the X-ray structure of a "PETase" from *Iodanella sakaiensis* at 1.5 Angstrom resolution. This bacterium was discovered recently by a Japanese team and they showed in an excellent publication in the journal "Science" that this strain produces the secreted PETase, which is able to hydrolyze polyethyleneterephthalate (PET) into BHET and MHET. These degradation products are then taken up by the bacterium, which also produces an intracellular so-called MHETase, which hydrolyses this to terephthalic acid (TA) and ethylene glycol (see Fig. 3 in the Yoshida et al. paper). This is important to mention, as in the manuscript by Joo et al., the substrates for the PETase are not correctly described!

The *Iodanella* strain then also has the enzymes available to assimilate TA completely and hence can grow on PET as carbon source.

The Japanese team already showed in their Science paper the relationship of the PETase to other hydrolases such as cutinase (see especially Table S2 in the supporting information) and they could show that it is a typical carboxylesterase. The Japanese team already showed in their paper (Figure S4) the residues of the catalytic triad of the PETase as well as of the oxyanion hole. This is important to state as Joo et al. now elaborate about their discovery of these residues, which is wrong!

Nevertheless, it is very important to have the 3D structure of the PETase now available.

It would indeed be especially interesting to understand how this unique enzyme indeed can take a PET-fiber, accommodate it in the active site and release the degradation products (this is unfortunately not shown properly in this paper!). Overall, this manuscript would nicely fit into standard enzyme structure journals such as *Acta Cryst. D.* for the reasons given here.

Joo et al. claim that they provide important insights into the molecular mechanism of the PETase, but this is only partially true. First of all, it was already clear from the Yoshida-Science paper, that the PETase is a typical carboxylesterase for which the active site residues are known, the GX SXG motif was reported as well as the oxyanion hole residues and that it belongs to the alpha/beta hydrolase fold family. Consequently, it can be expected that the PETase mechanistically hydrolyzes esters like all common carboxylesterases and especially those with an alpha/beta hydrolase fold (this has now been confirmed by Joo et al.). I appreciate nevertheless that Joo et al. made point mutations of the active site residues and show that activity

is greatly reduced or gone (as to be expected).

Secondly, the most interesting question is unfortunately not addressed: how can the PET fiber be accommodated in the active site region? This is completely unclear so far and providing experimental evidence for this would indeed be of high interest to the scientific community. Possibly this PETase uses similar activation tricks like lipases (most enzymes studied for PET degradation are indeed lipases or the related cutinases) where interfacial activation is important (see first lipase structures with and without inhibitor, Brady et al., Brzozowski et al., citations are given below).

I have also missed a detailed comparison to mechanistic analysis of the cutinase/lipase enzymes known to degrade PET. The section on p. 12 gives some information, but to state "was quite similar" is too simple. On the other hand, if the cutinase/lipase enzymes have high homology and the mechanism is 'quite similar' (as I expect), then this manuscript lacks sufficient novelty/importance. For subsite II analysis see my comments below about the modelling quality using a by far too simplified substrate mimic.

Furthermore, I have missed citations of a range of important publications:

Important papers about this fold family: Ollis, D. L. et al. The α/β hydrolase fold. *Protein Eng.* 5, 197-211 (1992); Updates on this: Kourist, R., Jochens, H., Bartsch, S., Kuipers, R., Padhi, S.K., Gall, M., Böttcher, D., Joosten, H.-J., Bornscheuer, U.T. The α/β -hydrolase fold 3DM database (ABHDB) as a tool for protein engineering, *ChemBioChem*, 11, 1635-164 (2010); Rauwerdink, A. & Kazlauskas, R. J. How the Same Core Catalytic Machinery Catalyzes 17 Different Reactions: the Serine-Histidine-Aspartate Catalytic Triad of alpha/beta-Hydrolase Fold Enzymes. *ACS Catal* 5, 6153-6176, doi:10.1021/acscatal.5b01539 (2015).

Lipase structures with important mechanistic insights: Brady, L. et al. A serine protease triad forms the catalytic centre of a triacylglycerol lipase. *Nature* 343, 767-770 (1990); Brzozowski, A. M. et al. A model for interfacial activation in lipases from the structure of a fungal lipase-inhibitor complex. *Nature* 351, 491-494 (1991).

Further important papers related to the Yoshida et al. *Science* paper, which must be cited to complement the discovery of *Iodanella sakaiensis*:

Bornscheuer, U. T. Feeding on plastic. *Science* 351, 1154-1155, doi:10.1126/science.aaf2853 (2016); Yang, Y., Yang, J. & Jiang, L. Comment on "A bacterium that degrades and assimilates poly(ethylene terephthalate)". *Science* 353, 759, doi:10.1126/science.aaf8305 (2016); Yoshida, S. et al. Response to Comment on "A bacterium that degrades and assimilates poly(ethylene terephthalate)". *Science* 353, 759, doi:10.1126/science.aaf8625 (2016).

Further comments:

Abstract and main text: Joo et al. claim that the active site accommodates two MHET molecules, which I don't understand because the PETase must be able to accommodate the entire PET polymer and the cleavage site would then be the ester bond between ethylene glycol and TA.

Furthermore, as outlined above, they are wrong (as this is contrast to the Yoshida paper, see the Table S2 in supporting information of the Science paper) that PETase makes TA and ethylene glycol from MHET. Yoshida et al. published that the PETase has no activity on MHET (that is why the bacterium has the second enzyme, MHETase).

p.2, line 58: provide detailed information about the chemical methods for plastic degradation, especially for PET.

lines 61/62: only a few plastics are polyesters (or polyamides) and hence only those can be hydrolyzed. Hydrolysis does not work for e.g. PP, PE, PVC, PEEK, polystyrene etc.

They have not determined the structure of the wildtype PETase, but a version which has many additional residues (after they removed the signal peptide sequence). So 42 amino acids were added to the 270 amino acid PETase. I really wonder, how much this could have influenced the 3D structure determined.

Furthermore, I am VERY puzzled that they report disulfide bonds in the structure! They expressed the gene in the E.coli Rosetta strain. This strain has additional aminoacyl tRNAs to assist expression of genes with uncommon codons, but does not has the ability to make disulfide bonds (efficiently). For this special E. coli strains are needed (see for instance: Stewart et al., EMBO Journal Vol.17 No.19 pp.5543–5550, 1998 and DOI: 10.1186/1475-2859-8-26)

p.7 line 126: the molecule mimicking PET is a very simply choice as it has little to do with the properties of the polymer in terms of size, accessible surface, hydrophobicity etc. This could be used for a first (inaccurate) modelling to reveal hints, but the real PET polymer (or at least a 10-20-mer) must be used here.

p. 10, PET degradation mechanism: this is more or less text book knowledge how esterase (and lipase) perform hydrolysis of ester bonds. Nothing new or unexpected described here.

p. 11/12: the experiments using PET film nicely confirm what Yoshida et al. have published. But nothing is new here.

Methods: the nucleotide sequence of the codon-optimized gene is missing. Moreover, I wonder what deviates in their protocol from the reported one by Yoshida et al. for expression and purification. In turn, if it differs, why did they use a different protocol than Yoshida et al.?

Reviewer #2 (Remarks to the Author):

The experimental work is suitable to be published, but I like to request modifications in the text, mainly for the section of the catalytic mechanism.

line 203 - 211

The charge-relay mechanism is not an up-to-date explanation. The activation of the catalytic serine relies on the polarization by hydrogen bonds in the catalytic triad. This makes the Ser-hydroxyl group more nucleophilic to attack the carbonyl-C of the substrate. A new ester (with the enzyme) is formed which is in a second step hydrolysed (a water molecule is required).

line 205 "A carbonyl oxygen is than forced to accept an electron, ..." this sounds like undergrad

chemistry (remove, or at least modify this).

line 207, the oxyanion hole is formed by ... and the tetrahedral intermediate is not collapsing, its formation is supported by the oxyanion hole. It is a transition state, short living !!! the cleavage takes place immediately.

line 208, not breakage, cleavage sounds more chemical.

* In general, the authors should not discuss the mechanism in detail, because the mechanism of this type of enzymes (Ser-hydrolases) is well known.*

line 261/2 and 262 "the only exception is Tyr87, ..." No, it is not an exception, because is is replaced by Phe.

line 406 ... 47,00% it is an approximation (line 405), don't give such exact value.

line 582 the Wilson B-factor is not subject of refinement, but belongs more to the data collection part.

This table is suitable for the supplement.

Figure 2a shows the standard catalytic site for Ser-hydrolases - delete and make fig. 2b larger.

In Fig. 4 or elsewhere: the chemical structure of the substrate with hydrogens, double bonds and aromatic moieties should be shown. This makes the figure more inviting to think about biotechnological chances.

--- end of comments ---

Reviewer #3 (Remarks to the Author):

This manuscript by Joo et al. presents the crystal structure of poly(ethylene terephthalate) esterase, and enzyme that hydrolyses this plastic polymer. As such, this represents an important enzyme for the management of the ever-increasing issue of accumulating plastic wastes. An understanding of this enzyme may help with the development of an efficient biodegradation process, and thereby greatly alleviate the problem of waste management we currently face worldwide. Hence, this study has the potential of great significance. The authors solved the structure of the enzyme with relatively high resolution and present a phylogenetic study that may serve to identify other esterases with similar substrate specificity. Unfortunately, however, by failing to obtain a structure of the esterase in complex with a ligand, the authors had to rely on a molecular docking model to identify potentially important residues for activity beyond the recognizable catalytic triad of Ser-His-Asp. Moreover, supporting kinetic data are lacking leaving much of the interpretation as only speculation. These issues are described in further detail below.

1. Line 116: The authors state that all three Ser-His-Asp residues function as covalent

nucleophiles - but I believe they mean to suggest that all three form a catalytic triad to render the Ser nucleophilic which could serve as the catalytic nucleophile. Nonetheless, without having yet presented the SDM data, this can only be suggested or predicted at this juncture.

2. The structure of the enzyme is generally very similar to other Ser hydrolases and there is nothing presented that distinguishes it from these others. That it likely functions as a Ser esterase could be readily predicted based on sequence alignments as searches. Unfortunately, this lessens the significance of this work beyond confirming what would have been predicted.

3. Line 123: The authors tried to both soak and co-crystalize with a substrate mimic but failed to obtain the structure of a complex. They assumed poor affinity of binding. Was this tested for in binding assays? If not, why not? Are there other compounds available that could have been tested?

Unfortunately, without this, the authors resorted to molecular docking using a substrate mimic. Why this mimic?

4. The authors then proceed to describe the binding of this mimic in great detail using language that does not convey the hypothetical nature of the observations; the text reads as if this is so. This needs to be corrected.

5. Lines 198-211: Based on this docking experiment, the authors then present a mechanism of action involving the catalytic triad. They present only the first half of the reaction, that involving the formation of the first transition state. Water is not mentioned, which would be involved in releasing the second product through a second transition state. Having said this, the mechanism would involve a covalent intermediate, and likely a ping-pong, bi-bi pathway. Is there any evidence for either? Certainly the latter can be obtained quite readily through a kinetic characterization.

6. Site-directed mutagenesis was performed to replace potentially important residues in order to predict their function. While it is unlikely that each of these replacements caused any folding issues, the authors should nonetheless have conducted an analysis to assure the reader, eg. Circular dichroism.

In general, despite the claim on line 90 that the detailed mechanism of the esterase is presented, unfortunately without much direct observation and kinetic data, the mechanism can only be proposed at this time. Likewise, the process for PET film degradation described in lines 237-245 was not demonstrated experimentally and so the text is only speculative.

Minor points:

Line 41: I think the authors meant “not” instead of “now”

Line 62: Only hydrolysis? Or do the authors mean enzyme activity, where hydrolysis is one reaction type?

Lines 156, 160, 167, 186 (and elsewhere?): “mutated” should read “replaced with” (as genes are mutated while amino acids are replaced).

Lines 157, 164, 168, 170, 173, 182, 189 (and elsewhere?): “variants” should be used to replace “mutants”

Lines 187: “replacing” instead of “mutating”

Line 191: the H bond is “predicted” (it was not observed)

Line 220: Nothing was truly observed, the authors are predicting or proposing.

Reviewer #4 (Remarks to the Author):

The paper describes the crystal structure, mechanism of action and structural relationships of a recently identified novel PET hydrolase from *Ideonella sakaiensis* (IsPETase). Various hydrolytic enzymes have been shown to cleave the ester bonds in the PET polymer, but their activity is rather low as PET is not a natural substrate of these enzymes. The PET hydrolase from *I. sakaiensis* is unique, as it has a natural role in PET degradation allowing the bacterial strain to use PET as a carbon source (Yoshida et al., 2016, *Science* 351, 1196). Therefore, its PET binding specificity and degrading activity are significantly higher compared to the other enzymes, which makes IsPETase highly attractive for biotechnological applications towards PET waste reduction and/or recycling. With the availability of its crystal structure, and the insights into the mechanism of substrate binding and cleavage, protein engineering efforts can now be focussed on further improving its enzymatic properties. In addition, evolutionary questions can be addressed how enzymes evolve to acquire new or improved activities. Thus, the research and results described in this paper provide a significant advance towards tackling a huge environmental problem, as well as allowing a better theoretical understanding of how (rapid) natural adaption of enzymes may take place.

The research described in the paper is overall sound and straightforward. The results concerning the PET degradation mechanism, based on the crystal structure and mutagenesis results, are

convincing. My main overall objection concerns the molecular docking procedure and the reliability of the docking results. Reliable docking is not trivial, and the authors do not specify the criteria they use to select the best binding pose for each modelled substrate. Is the O- γ atom of Ser-160 at a proper distance from the carbonyl carbon atom in the scissile ester bond in accordance with its role as nucleophile in the catalytic mechanism of the enzyme? What is the distance of the carbonyl oxygen atom in the scissile ester bond relative to main chain amide nitrogens of residues Met161 and Tyr87, forming the oxyanion hole? And did the authors use flexible docking, allowing some movement of side chains in the binding pocket? Arguably, a better approach (and used by others in similar scenarios, e.g., Juhl et al., 2009, BMC Structural Biology 9:39) would be to covalently dock the PET-like substrate to the enzyme in its tetrahedral intermediate state and improve the structure further by molecular mechanics/dynamics. This would also strengthen the conclusions drawn from the structural comparisons and phylogenetic tree analysis.

In addition I have a few other comments that should be addressed by the authors:

- The paper contains several typo errors and grammatical mistakes. This should be carefully checked (perhaps by a native speaker?). Also, the paper is a bit lengthy. In my opinion it can be reduced by carefully moving some information to the Supplemental section, and/or shortening some of the sections (e.g, the sections “PET degradation mechanism by IsPETase” and “Structural comparison of IsPETase with other PET degrading enzymes” may be substantially shortened without losing content)
- Line 41: “now” change to “not”
- Lines 97-99” The authors should specify what are the additional amino acid residues in the construct (in methods or supplemental section). I assume that the extra residues at the N-terminus contain a His-tag and thrombin-cleavage site?
- Lines 104-106: space group P212121 does not contain any pure 2-fold rotation axis (only screw axes), thus –by definition- it is not possible to generate a dimer via crystallographic symmetry. In other words, the part stating “and there was no symmetry operation” can be deleted from the sentence.
- Line 123: is the binding affinity of BHET known, or can it be measured?
- Lines 130-153: see main comment above. Is the scissile ester bond properly oriented with respect to the catalytically important residues (Ser160, Met161, Tyr87)? How did the authors address possible flexibility in the substrate binding pocket?

- Lines 256-257: I guess that the root-mean-square-deviations refer to C α -backbones only. This should be specified in the text.

- Lines 304-311: The conditions of the thermal stability assay are not mentioned in the paper (not in the methods section, nor in the supplemental part). In particular it would be necessary to know the pH at which the assay was conducted. It strikes me that the in-vitro catalytic assay with PET is carried out at pH 9, while the catalytic assay with BHET is carried out at pH 7. Thus, it would be crucial to measure the T_m-values of IsPETase wild-type and mutants at pH 9, or both pH 7 and pH 9.

- Line 340: “seem to have lower PET-degrading activities” “are predicted to have lower PET-degrading activities”

- Lines 405-406: remove “approximately” (2x)

- Lines 413-423: The description of the molecular docking procedure should be improved, or the procedure itself should be improved (see earlier comments).

“Auto dock Vina” change to “AutoDock Vina”.

“theoretical affinity of the binding” change to “calculated free energy of binding”.

Did the authors use flexible docking? How many poses were calculated and how were poses ranked? Was the final selected pose also the pose with the lowest free energy of binding? How did the authors validate the docking results?

- Lines 431 and 439: The authors should specify the time period for the reaction incubations.

- Figure 2A: The triad is not correctly modelled. The side chain of H237 should be 180 degrees rotated such that S160 can make a H-bond with N ϵ 2 and D206 with N δ 1.

- Figure 2C: The stereo-picture should be improved (the stereo-effect is not properly generated, possibly because the rotational difference between the two pictures is less than 6 degrees)

- It would help if a figure is added (in the main paper or as a supplemental figure) showing the chemical structures of the various substrates mentioned in the paper.

Response to Editor's and Reviewers' Comments

Manuscript ID: NCOMMS-17-13483-T

Editor's Comments to Author:

Your manuscript entitled "Structural insight into molecular mechanism of poly(ethylene terephthalate) degradation" has now been seen by 4 referees. You will see from their comments below that while they find your work of interest, some important points are raised. We are interested in the possibility of publishing your study in Nature Communications, but would like to consider your response to these concerns in the form of a revised manuscript before we make a final decision on publication. We therefore invite you to revise and resubmit your manuscript, taking into account the points raised. In particular we think it would be important to provide further experimental support for the proposed mechanism. Please highlight all changes in the manuscript text file.

[RESPONSE] We thank your and the reviewers' comments, which were invaluable for improving our manuscript. We have now addressed all the reviewers' comments with additional experiments and clearer description of some results, which are detailed below. Most importantly, we performed covalent docking calculation using longer PET substrate as reviewers suggested. With these new docking results, we were able to newly identify more extended substrate binding sites that can accommodate four MHET moieties of the PET polymer, which became one of the highlights of this work. Based on the newly identified substrate binding sites, we additionally performed protein engineering experiment as follows. We found that Arg280 was protruding within the newly identified substrate binding site, which led us to think that it might hinder substrate binding. Also, Arg is a polar amino acid. Thus, this residue (Arg280) was changed to Ala280 to remove its protruding side chain and make it non-polar, hoping that substrate binding will be improved. As we hypothesized, the engineered *Is*PETase having a replaced residue (Ala280) showed enhanced PETase activity. We solved the crystal structure of this variant (Arg280Ala) as well, which showed that the structure was changed to better accommodate the PET substrate as we hypothesized. This is important as the structure-based engineering of a residue (Arg280) which is located far away from the catalytic site with a distance of ~ 23 Å could be selected for enhancing the PETase activity. Taken together, the results presented in our original manuscript and also additional results reported in this revision now disclose the structure-based detailed mechanisms of this

unique PET degrading enzyme, *IsPETase*. Also, the proof-of-concept protein engineering demonstrated in this paper based on the 3D structure of *IsPETase* will be invaluable for further rational protein engineering. We detailed point-by-point responses to the reviewers' comments below.

Reviewers' Comments to Author:

Reviewer #1:

In this paper Joo et al. describe the X-ray structure of a "PETase" from *Iodanella sakaiensis* at 1.5 Angstrom resolution. This bacterium was discovered recently by a Japanese team and they showed in an excellent publication in the journal "Science" that this strain produces the secreted PETase, which is able to hydrolyze polyethyleneterephthalate (PET) into BHET and MHET. These degradation products are then taken up by the bacterium, which also produces an intracellular so-called MHETase, which hydrolyses this to terephthalic acid (TA) and ethylene glycol (see Fig. 3 in the Yoshida et al. paper). This is important to mention, as in the manuscript by Joo et al., the substrates for the PETase are not correctly described!

The *Iodanella* strain then also has the enzymes available to assimilate TA completely and hence can grow on PET as carbon source.

[RESPONSE] We agree that a couple of sentences we wrote in our original manuscript were confusing. As the reviewer described, *IsPETase* hydrolyzes PET into MHET (major product), BHET and terephthalate (TPA), and then MHET is further degraded into TPA and ethylene glycol (EG) by MHETase (Fig. 3 in the Yoshida et al. paper). This is what we also described by our own confirmation experiments and also by citing Yoshida et al.'s work. In our original manuscript, we previously wrote "These results suggest that *IsPETase* possesses strong degradation activity from BHET to MHET and EG, but **only ignorable** activity for further degradation of MHET to TPA and EG." However, we also wrote "In *I. sakaiensis*, PET is degraded by *IsPETase* to mono(2-hydroxyethyl) terephthalate (MHET), and then to TPA and EG mainly by MHETase, and **some by** *IsPETase*." In the above sentence of the original manuscript, "some by *IsPETase*" should have been written as "ignorable extent by *IsPETase*". We revised the entire manuscript stating that *IsPETase* has no activity on MHET, which was what we experimentally confirmed in the original manuscript as Yoshida et al. reported before.

The Japanese team already showed in their Science paper the relationship of the PETase to

other hydrolases such as cutinase (see especially Table S2 in the supporting information) and they could show that it is a typical carboxylesterase. The Japanese team already showed in their paper (Figure S4) the residues of the catalytic triad of the PETase as well as of the oxyanion hole. This is important to state as Joo et al. now elaborate about their discovery of these residues, which is wrong!

[RESPONSE] We fully acknowledged the great work by Yoshida et al. Throughout our manuscript, their work has been cited whenever needed. The Japanese team showed the relationship of *Is*PETase to other hydrolases previously, and suggested the catalytic triad and the oxyanion hole based on sequence alignment and analyses. We also agree that the overall fold of the enzyme and catalytic triad can be easily speculated. However, the most important findings and results we report are: actual crystal structure of this interesting enzyme, the unique conformation of the substrate binding site which is completely different from known cutinases, and detailed reaction mechanisms. The unique conformation of *Is*PETase and detailed reaction mechanisms including very interesting substrate binding scheme cannot be speculated/suggested without determination of the 3D structure. Also, thanks to this and other reviewers' comments, we further strengthened our paper through additional experiments providing new information on substrate binding and also protein engineering in the revised manuscript.

Nevertheless, it is very important to have the 3D structure of the PETase now available.

It would indeed be especially interesting to understand how this unique enzyme indeed can take a PET-fiber, accommodate it in the active site and release the degradation products (this is unfortunately not shown properly in this paper!). Overall, this manuscript would nicely fit into standard enzyme structure journals such as *Acta Cryst. D.* for the reasons given here.

[RESPONSE] We believe that the detailed 3D structure of *Is*PETase together with experimentally validated unique conformation of the substrate binding site and detailed reaction mechanisms are very important and of broad interest most suitable for publication in *Nature Communications*.

Joo et al claim that they provide important insights into the molecular mechanism of the PETase, but this is only partially true. First of all, it was already clear from the Yoshida-Science paper, that the PETase is a typical carboxylesterase for which the active site residues are known, the GX SXG motif was reported as well as the oxyanion hole residues and that it

belongs to the alpha/beta hydrolase fold family. Consequently, it can be expected that the PETase mechanistically hydrolyzes esters like all common carboxylesterases and especially those with an alpha/beta hydrolase fold (this has now been confirmed by Joo et al.). I appreciate nevertheless that Joo et al. made point mutations of the active site residues and show that activity is greatly reduced or gone (as to be expected).

[RESPONSE] Yes, the sequence information of *IsPETase* already suggests what this reviewer commented above. As described above, however, our main objective of research was why *IsPETase* shows, very interestingly, predominant activity for degrading PET differently from other cutinases. Although the fold and catalytic mechanism of *IsPETase* are similar with other carboxylesterases, we were able to suggest based on our structural studies that the differences in its substrate binding site are the most important feature for the strong PET degrading activity of *IsPETase*. Also, detailed results on reaction mechanisms and substrate binding (in particular thorough additional experiments during the revision) provide important information on the characteristics of *IsPETase* in its PET degradation.

Secondly, the most interesting question is unfortunately not addressed: how can the PET fiber be accommodated in the active site region? This is completely unclear so far and providing experimental evidence for this would indeed be of high interest to the scientific community. Possibly this PETase uses similar activation tricks like lipases (most enzymes studied for PET degradation are indeed lipases or the related cutinases) where interfacial activation is important (see first lipase structures with and without inhibitor, Brady et al., Brzozowski et al., citations are given below).

I have also missed a detailed comparison to mechanistic analysis of the cutinase/lipase enzymes known to degrade PET. The section on p. 12 gives some information, but to state "was quite similar" is too simple. On the other hand, if the cutinase/lipase enzymes have high homology and the mechanism is 'quite similar' (as I expect), then this manuscript lacks sufficient novelty/importance. For subsite II analysis see my comments below about the modelling quality using a by far too simplified substrate mimic.

[RESPONSE] Thank you for the great comment. As the reviewer suggested, the accommodation of PET substrate by PETase is very important to understand the mechanism of PET degradation. During this revision, we performed new docking experiment (as suggested by other reviewers). We were able to suggest that *IsPETase* uses a flat hydrophobic surface with dimensions of approximately 25 and 29 Å. The substrate binding site is located

on the flat surface and forms a long, shallow L-shaped cleft (new Fig.2). The detailed results are described in the revised manuscript. We are not sure whether this type of binding can be related to interfacial activation, but this binding seems to be important for initial contact between the enzyme and PET substrate.

Regarding the “cutinase/lipase” story, other reviewers suggested to cut down the description, and thus we did as they suggested.

Furthermore, I have missed citations of a range of important publications:

Important papers about this fold family: Ollis, D. L. et al. The α/β hydrolase fold. *Protein Eng.* 5, 197-211 (1992); Updates on this: Kourist, R., Jochens, H., Bartsch, S., Kuipers, R., Padhi, S.K., Gall, M., Böttcher, D., Joosten, H.-J., Bornscheuer, U.T. The α/β -hydrolase fold 3DM database (ABHDB) as a tool for protein engineering, *ChemBioChem*, 11, 1635-164 (2010); Rauwerdink, A. & Kazlauskas, R. J. How the Same Core Catalytic Machinery Catalyzes 17 Different Reactions: the Serine-Histidine-Aspartate Catalytic Triad of alpha/beta-Hydrolase Fold Enzymes. *ACS Catal* 5, 6153-6176, doi:10.1021/acscatal.5b01539 (2015).

Lipase structures with important mechanistic insights: Brady, L. et al. A serine protease triad forms the catalytic centre of a triacylglycerol lipase. *Nature* 343, 767-770 (1990); Brzozowski, A. M. et al. A model for interfacial activation in lipases from the structure of a fungal lipase-inhibitor complex. *Nature* 351, 491-494 (1991).

Further important papers related to the Yoshida et al. *Science* paper, which must be cited to complement the discovery of *Iodanella sakaiensis*:

Bornscheuer, U. T. Feeding on plastic. *Science* 351, 1154-1155, doi:10.1126/science.aaf2853 (2016); Yang, Y., Yang, J. & Jiang, L. Comment on "A bacterium that degrades and assimilates poly(ethylene terephthalate)". *Science* 353, 759, doi:10.1126/science.aaf8305 (2016); Yoshida, S. et al. Response to Comment on "A bacterium that degrades and assimilates poly(ethylene terephthalate)". *Science* 353, 759, doi:10.1126/science.aaf8625 (2016).

[RESPONSE] In the original manuscript, we already cited all the important references. However, we honored this reviewer’s suggestion and added the references when the context is relevant in the revised manuscript.

Further comments:

Abstract and main text: Joo et al. claim that the active site accommodates two MHET

molecules, which I don't understand because the PETase must be able to accommodate the entire PET polymer and the cleavage site would then be the ester bond between ethylene glycol and TA. Furthermore, as outlined above, they are wrong (as this is contrast to the Yoshida paper, see the Table S2 in supporting information of the Science paper) that PETase makes TA and ethylene glycol from MHET. Yoshida et al. published that the PETase has no activity on MHET (that is why the bacterium has the second enzyme, MHETase).

[RESPONSE] As we already responded above, we clarified this point by rewriting the corresponding sentences.

p.2, line 58: provide detailed information about the chemical methods for plastic degradation, especially for PET.

[RESPONSE] We revised the sentence as follows:

“To remove plastic wastes and recycle plastic-based materials, several chemical degradation methods such as glycolysis, methanolysis, hydrolysis, aminolysis and ammonolysis have been developed³.”

lines 61/62: only a few plastics are polyesters (or polyamides) and hence only those can be hydrolyzed. Hydrolysis does not work for e.g. PP, PE, PVC, PEEK, polystyrene etc.

[RESPONSE] Thank you. We revised the sentence as follows:

“Microbes can degrade plastics with ester bond via enzymatic hydrolysis through colonization onto the surfaces of materials.”

They have not determined the structure of the wildtype PETase, but a version which has many additional residues (after they removed the signal peptide sequence). So 42 amino acids were added to the 270 amino acid PETase. I really wonder, how much this could have influenced the 3D structure determined.

[RESPONSE] In our current 3D structure, there are no electron density map for the additional residues, indicating that these residues are disordered and do not interact with the main protein. Thus, the influence of the additional residues to the 3D structure is safely ignored.

Furthermore, I am VERY puzzled that they report disulfide bonds in the structure! They expressed the gene in the E. coli Rosetta strain. This strain has additional aminoacyl tRNAs

to assist expression of genes with uncommon codons, but does not has the ability to make disulfide bonds (efficiently). For this special E. coli strains are needed (see for instance: Stewart et al., EMBO Journal Vol.17 No.19 pp.5543–5550, 1998 and DOI: 10.1186/1475-2859-8-26)

[RESPONSE] The reviewer seems to be completely confused. The strain we used for *IsPETase* expression was *E. coli* Rosetta gami-B, not *E. coli* Rosetta. The genotype of *E. coli* Rosetta gami-B is F⁻ *ompT hsdSB* (rB⁻ mB⁻) *gal dcm lacY1 ahpC* (DE3) *gor522::Tn10 trxB* pRARE (Cam^R, Kan^R, Tet^R). We used this strain exactly because of disulfide bond formation; this strain has *trxB/gor* mutations to improve formation of disulfide bonds in the cytoplasm of *E. coli* (Bessette et al., (1999) *Proc. Natl. Acad. Sci. U.S.A.* 96(24): 13703-13708). Moreover, as generally well known, disulfide bonds can also be formed *in vitro* in the presence of oxygen.

p.7 line 126: the molecule mimicking PET is a very simply choice as it has little to do with the properties of the polymer in terms of size, accessible surface, hydrophobicity etc. This could be used for a first (inaccurate) modelling to reveal hints, but the real PET polymer (or at least a 10-20-mer) must be used here.

[RESPONSE] We also agree that two MHET molecules are quite short to represent PET polymer. Based on this and other reviewers' comments, we newly performed docking calculation using 4 MHET molecules, and the chemical fits well into the substrate binding site (new Fig. 2). Although it is possible to bind more MHET molecules to the enzyme, we believe our suggestion is the best description we can draw from our structure and docking calculation. Based on the structure we determined and docking simulations, *IsPETase* does not bind more than 4 MHET. Interestingly, we found that this seems to be due to the presence of protruding Arg280 residue which is located near the last MHET moiety of 4-MHET substrate. For the interaction between *IsPETase* and PET, we propose that 4 MHET moieties are the most properly matched substrate due to cleft on structure even with the 10-20-mers for PET. We thank this and other reviewers for suggesting this important point, which led us to perform new additional experiments (including the follow-up protein engineering experiment) for providing new insights on PETase.

p. 10, PET degradation mechanism: this is more or less text book knowledge how esterase

(and lipase) perform hydrolysis of ester bonds. Nothing new or unexpected described here.

[RESPONSE] As the reviewer mentioned, the reaction mechanism of esterase is well known. However, this paper focuses on how the substrate binds to the enzyme and which differences in enzyme structure result in significantly higher PET degrading activity compared with other cutinases and esterases, which makes *IsPETase* highly attractive for industrial applications towards PET waste recycling. Based on the 3D structure and related biochemical studies, we can explain the reasons for extraordinary PET degrading activity of *IsPETase*. Thus, we cannot agree with this reviewer saying “nothing new”; through determination of the 3D structure of *IsPETase*, we report many unique interesting features with respect to substrate binding and reaction mechanisms.

p. 11/12: the experiments using PET film nicely confirm what Yoshida et al. have published. But nothing is new here.

[RESPONSE] This is scientifically unfair statement. Although their experiments using PET film show many important findings, our experiments are based on the FIRST detailed structural studies. Without structural studies, all the results described on mechanisms are just hypotheses (and they even did not provide hypotheses on detailed enzyme mechanisms). Most importantly, it was not possible to understand detailed conformation of the substrate binding site. Based on our structural analysis, we selected the key residues and demonstrated how these residues are involved in the formation of the substrate binding site. Importantly, we were able to make a variant (Arg280Ala) having higher PET degradation activity based on our structural information, which would be very difficult, if not impossible, to achieve. Thus, there are many important new findings in our paper.

Methods: the nucleotide sequence of the codon-optimized gene is missing. Moreover, I wonder what deviates in their protocol from the reported one by Yoshida et al. for expression and purification. In turn, if it differs, why did they use a different protocol than Yoshida et al.?

[RESPONSE] We added the sequence of the codon-optimized gene to Supplementary Table 2. Regarding the expression system, we tested several different vectors and host strains, and we chose the best system we described in the Methods section; we have enough experiences and knowhows over the years on protein expression and purification.

Reviewer #2:

The experimental work is suitable to be published, but I like to request modifications in the text, mainly for the section of the catalytic mechanism.

[RESPONSE] Thank you very much. We responded to all your comments below.

line 203 - 211 The charge-relay mechanism is not an up-to-date explanation. The activation of the catalytic serine relies on the polarization by hydrogen bonds in the catalytic triad. This makes the Ser-hydroxyl group more nucleophilic to attack the carbonyl-C of the substrate. A new ester (with the enzyme) is formed which is in a second step hydrolysed (a water molecule is required).

[RESPONSE] We agree to the reviewer's comments. However, during this revision, the "PET degradation mechanism by *IsPETase*" section describing catalytic mechanism was simplified as Reviewer 4 suggested. Thus, the explanation was deleted in revised manuscript.

line 205 "A carbonyl oxygen is than forced to accept an electron, ..." this sounds like undergrad chemistry (remove, or at least modify this).

[RESPONSE] Thank you. We removed the sentence.

line 207, the oxyanion hole is formed by ... and the tetrahedral intermediate is not collapsing, its formation is supported by the oxyanion hole. It is a transition state, short living !!! the cleavage takes place immediately.

[RESPONSE] We revised the sentence as follows:

"Oxyanion of the tetrahedral intermediate is stabilized by an oxyanion hole that consists of nitrogen atoms of Tyr87 and Met160 with distances of 2.90 Å and 2.83 Å, respectively (**Fig. 2a**)."

line 208, not breakage, cleavage sounds more chemical. * In general, the authors should not discuss the mechanism in detail, because the mechanism of this type of enzymes (Ser-hydrolases) is well known.*

[RESPONSE] We changed "breakage" to "cleavage", and removed explanation on the mechanism as suggested.

line 261/2 and 262 "the only exception is Tyr87, ..." No, it is not an exception, because is is replaced by Phe.

[RESPONSE] We removed the sentence.

line 406 ... 47,00% it is an approximation (line 405), don't give such exact value.

[RESPONSE] Thank you for the comment. Each value for the Matthews coefficient and solvent content is changed by using exact value of molecular weight of *IsPETase*. We revised the sentence as follows:

“With one molecule of *IsPETase* per asymmetric unit, the Matthews coefficient was 2.64 Å³·Da⁻¹, which corresponds to a solvent content of 53.38%³³”

line 582 the Wilson B-factor is not subject of refinement, but belongs more to the data collection part. This table is suitable for the supplement.

[RESPONSE] Thank you. We moved the table and Wilson B factor to new Supplementary Table 1 and data collection part, respectively.

Figure 2a shows the standard catalytic site for Ser-hydrolases - delete and make fig. 2b larger. In Fig. 4 or elsewhere: the chemical structure of the substrate with hydrogens, double bonds and aromatic moieties should be shown. This makes the figure more inviting to think about biotechnological chances.

[RESPONSE] We redrew Fig. 2 based on the new docking results considering the reviewer's comment. We also made Supplementary Fig. 3 showing chemical structure of the substrates.

Reviewer #3:

This manuscript by Joo et al. presents the crystal structure of poly(ethylene terephthalate) esterase, and enzyme that hydrolyses this plastic polymer. As such, this represents an important enzyme for the management of the ever-increasing issue of accumulating plastic wastes. An understanding of this enzyme may help with the development of an efficient biodegradation process, and thereby greatly alleviate the problem of waste management we currently face worldwide. Hence, this study has the potential of great significance. The authors solved the structure of the enzyme with relatively high resolution and present a phylogenetic study that may serve to identify other esterases with similar substrate specificity.

[RESPONSE] Thank you for recognizing the importance of this work.

Unfortunately, however, by failing to obtain a structure of the esterase in complex with a

ligand, the authors had to rely on a molecular docking model to identify potentially important residues for activity beyond the recognizable catalytic triad of Ser-His-Asp. Moreover, supporting kinetic data are lacking leaving much of the interpretation as only speculation. These issues are described in further detail below.

1. Line 116: The authors state that all three Ser-His-Asp residues function as covalent nucleophiles - but I believe they mean to suggest that all three form a catalytic triad to render the Ser nucleophilic which could serve as the catalytic nucleophile. Nonetheless, without having yet presented the SDM data, this can only be suggested or predicted at this juncture.

[RESPONSE] We also intended to describe Ser as the catalytic nucleophile and we presented the SDM data for several important residues including the three Ser-His-Asp residues in Fig. 3a. To clearly describe, we revised the sentences as follows:

“At the active site of *Is*PETase, three residues Ser160, His237, and Asp206 form a catalytic triad and Ser160 functions as a covalent nucleophile to the carbonyl carbon atom in the scissile ester bond, as in other carboxylesterases (**Fig. 2a**).”

2. The structure of the enzyme is generally very similar to other Ser hydrolases and there is nothing presented that distinguishes it from these others. That it likely functions as a Ser esterase could be readily predicted based on sequence alignments as searches. Unfortunately, this lessens the significance of this work beyond confirming what would have been predicted.

[RESPONSE] The major significance of our work is understanding how PET substrate is accommodated to *Is*PETase with the distinct substrate binding site resulting in superior PET degrading activity of this enzyme. As described in “Structural comparison of *Is*PETase with other PET degrading enzymes” section, we showed that *Is*PETase has unique structural features, especially on the substrate binding site. Such finding (along with other findings on mechanisms) cannot be obtained by prediction based on sequence alignments only as in Yoshida et al’s work.

3. Line 123: The authors tried to both soak and co-crystallize with a substrate mimic but failed to obtain the structure of a complex. They assumed poor affinity of binding. Was this tested for in binding assays? If not, why not? Are there other compounds available that could have been tested? Unfortunately, without this, the authors resorted to molecular docking using a substrate mimic. Why this mimic?

[RESPONSE] To obtain the crystal structure of *Is*PETase with substrate, we used only BHET

because there are no commercially available chemicals except BHET. Although we tried to measure the kinetic data of *IsPETase* using BHET, we failed to obtain reliable data due to the low solubility of BHET. Because we are not sure whether the affinity for BHET is low or not, we revised the sentence as follows:

“...potentially because we could not use high concentration of BHET in co-crystallization and soaking due to its low solubility.”

During this revision, we performed new docking calculation with longer substrate as suggested by the reviewers. At the end, we were able to make the docking complex of 4 MHET moieties and *IsPETase*.

4. The authors then proceed to describe the binding of this mimic in great detail using language that does not convey the hypothetical nature of the observations; the text reads as if this is so. This needs to be corrected.

[RESPONSE] Thank you. We revised the “Active site of *IsPETase*” section according to the reviewer’s advice.

5. Lines 198-211: Based on this docking experiment, the authors then present a mechanism of action involving the catalytic triad. They present only the first half of the reaction, that involving the formation of the first transition state. Water is not mentioned, which would be involved in releasing the second product through a second transition state. Having said this, the mechanism would involve a covalent intermediate, and likely a ping-pong, bi-bi pathway. Is there any evidence for either? Certainly the latter can be obtained quite readily through a kinetic characterization.

[RESPONSE] Since the catalytic mechanism involving the catalytic triad is already well known, we did not describe the catalytic process in detail (Reviewer 4 suggested to shorten the section). It is also well known that hydrolase enzymes have ping-pong bi-bi mechanism (Ringborg et al., (2016) *React. Chem. Eng.* 1: 10-22). As we responded to comment 2, *IsPETase* has similar catalytic residues and overall enzyme fold to other hydrolases. Therefore, we conclude that *IsPETase* also has ping-pong bi-bi mechanism.

6. Site-directed mutagenesis was performed to replace potentially important residues in order to predict their function. While it is unlikely that each of these replacements caused any folding issues, the authors should nonetheless have conducted an analysis to assure the reader,

eg. Circular dichroism.

[RESPONSE] As the reviewer advised, we additionally performed CD experiments, and added the data to Supplementary Fig. 6. The CD spectra indicate that there are no significant differences among the variants and the wild-type *Is*PETase.

In general, despite the claim on line 90 that the detailed mechanism of the esterase is presented, unfortunately without much direct observation and kinetic data, the mechanism can only be proposed at this time. Likewise, the process for PET film degradation described in lines 237-245 was not demonstrated experimentally and so the text is only speculative.

[RESPONSE] We agree that PET degradation process still remains speculative. It is extremely difficult to obtain the kinetic data of PET hydrolysis. However, in structural perspective, we can narrow down the possible positioning of the substrate to propose the mechanism. Moreover, in the revision, we present newly constructed variant (Arg280Ala) showing enhanced PET degrading activity. This successful enzyme engineering supports the reliability of docking calculation and identification of substrate binding site.

Minor points:

Line 41: I think the authors meant “not” instead of “now”

[RESPONSE] Thank you. Corrected.

Line 62: Only hydrolysis? Or do the authors mean enzyme activity, where hydrolysis is one reaction type?

[RESPONSE] We changed “plastics” to “plastics with ester bond”.

Lines 156, 160, 167, 186 (and elsewhere?): “mutated” should read “replaced with” (as genes are mutated while amino acids are replaced).

[RESPONSE] We changed “mutated” to “replaced with”.

Lines 157, 164, 168, 170, 173, 182, 189 (and elsewhere?): “variants” should be used to replace “mutants”

[RESPONSE] We changed all “mutants” to “variants”.

Lines 187: “replacing” instead of “mutating”

[RESPONSE] We changed “mutating” to “replacing”.

Line 191: the H bond is “predicted” (it was not observed)

[RESPONSE] The H bond was removed in the revised manuscript.

Line 220: Nothing was truly observed, the authors are predicting or proposing.

[RESPONSE] Corrected as suggested.

Reviewer #4:

The paper describes the crystal structure, mechanism of action and structural relationships of a recently identified novel PET hydrolase from *Ideonella sakaiensis* (IsPETase). Various hydrolytic enzymes have been shown to cleave the ester bonds in the PET polymer, but their activity is rather low as PET is not a natural substrate of these enzymes. The PET hydrolase from *I. sakaiensis* is unique, as it has a natural role in PET degradation allowing the bacterial strain to use PET as a carbon source (Yoshida et al., 2016, *Science* 351, 1196). Therefore, its PET binding specificity and degrading activity are significantly higher compared to the other enzymes, which makes IsPETase highly attractive for biotechnological applications towards PET waste reduction and/or recycling. With the availability of its crystal structure, and the insights into the mechanism of substrate binding and cleavage, protein engineering efforts can now be focussed on further improving its enzymatic properties. In addition, evolutionary questions can be addressed how enzymes evolve to acquire new or improved activities. Thus, the research and results described in this paper provide a significant advance towards tackling a huge environmental problem, as well as allowing a better theoretical understanding of how (rapid) natural adaption of enzymes may take place.

[RESPONSE] Thank you very much for recognizing the importance of this work.

The research described in the paper is overall sound and straightforward. The results concerning the PET degradation mechanism, based on the crystal structure and mutagenesis results, are convincing. My main overall objection concerns the molecular docking procedure and the reliability of the docking results. Reliable docking is not trivial, and the authors do not specify the criteria they use to select the best binding pose for each modelled substrate. Is the O- γ atom of Ser-160 at a proper distance from the carbonyl carbon atom in the scissile ester bond in accordance with its role as nucleophile in the catalytic mechanism of the

enzyme? What is the distance of the carbonyl oxygen atom in the scissile ester bond relative to main chain amide nitrogens of residues Met161 and Tyr87, forming the oxyanion hole? And did the authors use flexible docking, allowing some movement of side chains in the binding pocket? Arguably, a better approach (and used by others in similar scenarios, e.g., Juhl et al., 2009, BMC Structural Biology 9:39) would be to covalently dock the PET-like substrate to the enzyme in its tetrahedral intermediate state and improve the structure further by molecular mechanics/dynamics. This would also strengthen the conclusions drawn from the structural comparisons and phylogenetic tree analysis.

[RESPONSE] Thank you for the constructive comments. Our previous docking models of BHET and 2-HE(MHET)₂ showed reasonable distance of 3.2 Å between the O_γ of Ser160 and the carbonyl carbon atom in the scissile ester. We used flexible docking and selected the best models of each compound by the calculated free energy of binding. We added the information for the selected flexible residues in the Methods section. The new docking approach the reviewer suggested is very helpful, but we do not have the license and the ability to deal with *FlexX* software in a short time. Instead, we performed mixed approach of induced-fit and covalent docking with the much longer ligand, 2-HE(MHET)₄ using AutoDock Vina and AutoDock, and the docking pose was minimized in the schrodinger suite (Glide). We obtained improved docking results from the covalent docking calculation, which strengthened the manuscript much more. We thank the reviewer for the invaluable comments again.

In addition I have a few other comments that should be addressed by the authors:

- The paper contains several typo errors and grammatical mistakes. This should be carefully checked (perhaps by a native speaker?). Also, the paper is a bit lengthy. In my opinion it can be reduced by carefully moving some information to the Supplemental section, and/or shortening some of the sections (e.g, the sections “PET degradation mechanism by IsPETase” and “Structural comparison of IsPETase with other PET degrading enzymes” may be substantially shortened without losing content)

[RESPONSE] Thank you. We checked the manuscript thoroughly and corrected typos. As the reviewer suggested, we also shortened two sections, “PET degradation mechanism by IsPETase” and “Structural comparison of IsPETase with other PET degrading enzymes”.

- Line 41: “now” change to “not”

[RESPONSE] We changed “now” to “not”

- Lines 97-99” The authors should specify what are the additional amino acid residues in the construct (in methods or supplemental section). I assume that the extra residues at the N-terminus contain a His-tag and thrombin-cleavage site?

[RESPONSE] As the reviewer noted, the additional residues at the N-terminus contain a His-tag and thrombin-cleavage site. We added the information on these residues to new Supplementary Table 2.

- Lines 104-106: space group P212121 does not contain any pure 2-fold rotation axis (only screw axes), thus –by definition- it is not possible to generate a dimer via crystallographic symmetry. In other words, the part stating “and there was no symmetry operation” can be deleted from the sentence.

[RESPONSE] Thank you. We deleted that sentence.

- Line 123: is the binding affinity of BHET known, or can it be measured?

[RESPONSE] Although we tried to measure the kinetic data of *Is*PETase, we failed to obtain reliable data due to the low solubility of BHET. Therefore, we revised the sentence as follows:

“...potentially because we could not use high concentration of BHET in co-crystallization and soaking due to its low solubility.”

- Lines 130-153: see main comment above. Is the scissile ester bond properly oriented with respect to the catalytically important residues (Ser160, Met161, Tyr87)? How did the authors address possible flexibility in the substrate binding pocket?

[RESPONSE] We responded in detail above. The docking models showed proper orientation and reasonable distance with respect to the residues Ser160, Met161 and Tyr87 (both original Fig. 2 and new Fig. 2). To address possible flexibility, we performed flexible docking calculation in which several flexible residues were selected. The docking calculation is described in detail on Methods section.

- Lines 256-257: I guess that the root-mean-square-deviations refer to C α -backbones only.

This should be specified in the text.

[RESPONSE] The root-mean-square-deviations refer to the backbone C α atomic coordinates. However, during this revision, the sentence was deleted to shorten the manuscript.

- Lines 304-311: The conditions of the thermal stability assay are not mentioned in the paper (not in the methods section, nor in the supplemental part). In particular it would be necessary to know the pH at which the assay was conducted. It strikes me that the in-vitro catalytic assay with PET is carried out at pH 9, while the catalytic assay with BHET is carried out at pH 7. Thus, it would be crucial to measure the T_m-values of IsPETase wild-type and mutants at pH 9, or both pH 7 and pH 9.

[RESPONSE] Thank you for the comment. We measured the T_m value of IsPETase at pH 7 and pH 9. The results are shown in new Supplementary Fig. 4.

- Line 340: “seem to have lower PET-degrading activities” “are predicted to have lower PET-degrading activities”

[RESPONSE] Thank you. We revised “seem to have lower PET-degrading activities” to “are predicted to have lower PET-degrading activities”

- Lines 405-406: remove “approximately” (2x)

[RESPONSE] We removed it.

- Lines 413-423: The description of the molecular docking procedure should be improved, or the procedure itself should be improved (see earlier comments).

[RESPONSE] We added the detailed description on docking calculation in Methods section.

“Auto dock Vina” change to “AutoDock Vina”.

“theoretical affinity of the binding” change to “calculated free energy of binding”.

[RESPONSE] Thank you. We changed “Auto dock Vina” to “AutoDock Vina” and “theoretical affinity of the binding” to “calculated free energy of binding” as the reviewer suggested.

Did the authors use flexible docking? How many poses were calculated and how were poses ranked? Was the final selected pose also the pose with the lowest free energy of binding? How did the authors validate the docking results?

[RESPONSE] As we newly described in Methods section, prior to the covalent docking, flexible docking, calculation using AutoDock Vina was performed, and nine output poses were generated with their calculated free energy of binding from its own scoring function. The best docking model with the lowest binding energy (-7.1 kcal/mol) was selected, and the conformation of the model was used as an evaluation standard for the following calculation.

- Lines 431 and 439: The authors should specify the time period for the reaction incubations.

[RESPONSE] We added the time period for incubation.

- Figure 2A: The triad is not correctly modelled. The side chain of H237 should be 180 degrees rotated such that S160 can make a H-bond with Nε2 and D206 with Nδ1.

[RESPONSE] Thank you. We revised Fig. 2 correctly.

- Figure 2C: The stereo-picture should be improved (the stereo-effect is not properly generated, possibly because the rotational difference between the two pictures is less than 6 degrees)

[RESPONSE] The stereo-picture is now deleted during the revision of the Fig. 2.

- It would help if a figure is added (in the main paper or as a supplemental figure) showing the chemical structures of the various substrates mentioned in the paper.

[RESPONSE] We made new Supplementary Fig. 3 showing the chemical structures of the substrates.

Reviewers' Comments:

Reviewer #2 (Remarks to the Author):

The modified manuscript is well improved according to all requests of the reviewers. I have seen the reviews, detailed responding text and the updated manuscript.

Therefore, my comment can be short with "acceptable manuscript".

Reviewer #3 (Remarks to the Author):

1. Line 116: The authors state that all three Ser-His-Asp residues function as covalent nucleophiles - but I believe they mean to suggest that all three form a catalytic triad to render the Ser nucleophilic which could serve as the catalytic nucleophile. Nonetheless, without having yet presented the SDM data, this can only be suggested or predicted at this juncture. [RESPONSE] We also intended to describe Ser as the catalytic nucleophile and we presented the SDM data for several important residues including the three Ser-His-Asp residues in Fig. 3a. To clearly describe, we revised the sentences as follows:

“At the active site of IsPETase, three residues Ser160, His237, and Asp206 form a catalytic triad and Ser160 functions as a covalent nucleophile to the carbonyl carbon atom in the scissile ester bond, as in other carboxylesterases (Fig. 2a).”

Yes, but without any kinetic data or the direct observation of a covalent adduct, you are still assuming the enzyme proceeds as stated. Hence, rather than stating as a fact that “Ser160 functions as a covalent nucleophile...” this should be tempered with saying that Ser160 is “assumed” or “postulated” to function in this manner.

2. The structure of the enzyme is generally very similar to other Ser hydrolases and there is nothing presented that distinguishes it from these others. That it likely functions as a Ser esterase could be readily predicted based on sequence alignments as searches. Unfortunately, this lessens the significance of this work beyond confirming what would have been predicted. [RESPONSE] The major significance of our work is understanding how PET substrate is accommodated to IsPETase with the distinct substrate binding site resulting in superior PET degrading activity of this enzyme. As described in “Structural comparison of IsPETase with other PET degrading enzymes” section, we showed that IsPETase has unique structural features, especially on the substrate binding site. Such finding (along with other findings on mechanisms) cannot be obtained by prediction based on sequence alignments only as in Yoshida et al’s work.

Yes, but again the complex was “predicted” based on in silico docking experiments and not

directly observed. Again, these predicted findings have been overstated and the tone of the discussion should reflect the predictive nature of the findings.

3. Line 123: The authors tried to both soak and co-crystallize with a substrate mimic but failed to obtain the structure of a complex. They assumed poor affinity of binding. Was this tested for in binding assays? If not, why not? Are there other compounds available that could have been tested? Unfortunately, without this, the authors resorted to molecular docking using a substrate mimic. Why this mimic? [RESPONSE] To obtain the crystal structure of IsPETase with substrate, we used only BHET because there are no commercially available chemicals except BHET. Although we tried to measure the kinetic data of IsPETase using BHET, we failed to obtain reliable data due to the low solubility of BHET. Because we are not sure whether the affinity for BHET is low or not, we revised the sentence as follows:

“...potentially because we could not use high concentration of BHET in cocrystallization and soaking due to its low solubility.”

During this revision, we performed new docking calculation with longer substrate as suggested by the reviewers. At the end, we were able to make the docking complex of 4 MHET moieties and IsPETase.

OK - but again affinity of BHET could be readily determined by a ligand binding study using, eg., SPR or ITC.

4. The authors then proceed to describe the binding of this mimic in great detail using language that does not convey the hypothetical nature of the observations; the text reads as if this is so. This needs to be corrected. [RESPONSE] Thank you. We revised the “Active site of IsPETase” section according to the reviewer’s advice.

5. Lines 198-211: Based on this docking experiment, the authors then present a mechanism of action involving the catalytic triad. They present only the first half of the reaction, that involving the formation of the first transition state. Water is not mentioned, which would be involved in releasing the second product through a second transition state. Having said this, the mechanism would involve a covalent intermediate, and likely a ping-pong, bi-bi pathway. Is there any evidence for either? Certainly the latter can be obtained quite readily through a kinetic characterization. [RESPONSE] Since the catalytic mechanism involving the catalytic triad is already well known, we did not describe the catalytic process in detail (Reviewer 4 suggested to shorten the section). It is also well known that hydrolase enzymes have ping-pong bi-bi mechanism (Ringborg et al., (2016) *React. Chem. Eng.* 1: 10-22). As we responded to comment 2, IsPETase has similar catalytic residues and overall enzyme fold to other hydrolases. Therefore, we conclude that IsPETase also has ping-pong bi-bi mechanism.

I understand these assumptions - but they are nonetheless assumptions. Certainly as suggested by Reviewer 4, the discussion of the mechanism could be abbreviated, but given the broader

readership of the Journal, it is not appropriate to ignore the second half of the reaction involving the addition of water to complete the hydrolytic reaction. This still needs to be included.

6. Site-directed mutagenesis was performed to replace potentially important residues in order to predict their function. While it is unlikely that each of these replacements caused any folding issues, the authors should nonetheless have conducted an analysis to assure the reader, eg. Circular dichroism. [RESPONSE] As the reviewer advised, we additionally performed CD experiments, and added the data to Supplementary Fig. 6. The CD spectra indicate that there are no significant differences among the variants and the wild-type IsPETase.

Thank you.

In general, despite the claim on line 90 that the detailed mechanism of the esterase is presented, unfortunately without much direct observation and kinetic data, the mechanism can only be proposed at this time. Likewise, the process for PET film degradation described in lines 237-245 was not demonstrated experimentally and so the text is only speculative. [RESPONSE] We agree that PET degradation process still remains speculative. It is extremely difficult to obtain the kinetic data of PET hydrolysis. However, in structural perspective, we can narrow down the possible positioning of the substrate to propose the mechanism. Moreover, in the revision, we present newly constructed variant (Arg280Ala) showing enhanced PET degrading activity. This successful enzyme engineering supports the reliability of docking calculation and identification of substrate binding site.

Nonetheless, my original concern still holds regarding the relatively strong language used to describe the mechanism; you still can only speculate or propose.

Minor points:

Line 41: I think the authors meant “not” instead of “now”

[RESPONSE] Thank you. Corrected.

Thank you

Line 62: Only hydrolysis? Or do the authors mean enzyme activity, where hydrolysis is one reaction type?

[RESPONSE] We changed “plastics” to “plastics with ester bond”.

Thank you

Lines 156, 160, 167, 186 (and elsewhere?): “mutated” should read “replaced with” (as genes are mutated while amino acids are replaced).

[RESPONSE] We changed “mutated” to “replaced with”.

Thank you

Lines 157, 164, 168, 170, 173, 182, 189 (and elsewhere?): “variants” should be used to replace “mutants”

[RESPONSE] We changed all “mutants” to “variants”.

Thank you

Lines 187: “replacing” instead of “mutating”

[RESPONSE] We changed “mutating” to “replacing”.

Thank you

Line 191: the H bond is “predicted” (it was not observed)

[RESPONSE] The H bond was removed in the revised manuscript.

OK

Line 220: Nothing was truly observed, the authors are predicting or proposing.

[RESPONSE] Corrected as suggested.

As noted above several times, the tone of the discussion is still too strong and it needs to reflect the predictive nature of the observations that are based on theoretical/in silico studies.

Reviewer #4 (Remarks to the Author):

The revised manuscript has improved considerably, and adequately addresses the main criticisms of the referees. In addition new results are included in the manuscript (e.g. docking of longer substrate, R280A mutant with enhanced PETase activity) which allow for a more thorough analysis and interesting discussion of the molecular basis of the enhanced PETase activity of this enzyme, as compared to its close homologs. Judged from the information presented in the manuscript the docking has been carried out carefully, and the results are highly relevant. In my view the manuscript is suitable for publication in Nature Comm.

Response to Reviewers' Comments

Manuscript ID: NCOMMS-17-13483A

Reviewer #2 (Remarks to the Author):

The modified manuscript is well improved according to all requests of the reviewers. I have seen the reviews, detailed responding text and the updated manuscript.

Therefore, my comment can be short with "acceptable manuscript".

[RESPONSE] Thank you very much for your invaluable comments on our original manuscript, which made our paper much more improved. Much appreciated.

Reviewer #3 (Remarks to the Author):

1. Line 116: The authors state that all three Ser-His-Asp residues function as covalent nucleophiles - but I believe they mean to suggest that all three form a catalytic triad to render the Ser nucleophilic which could serve as the catalytic nucleophile. Nonetheless, without having yet presented the SDM data, this can only be suggested or predicted at this juncture.

[RESPONSE] We also intended to describe Ser as the catalytic nucleophile and we presented the SDM data for several important residues including the three Ser-His-Asp residues in Fig. 3a. To clearly describe, we revised the sentences as follows:

“At the active site of IsPETase, three residues Ser160, His237, and Asp206 form a catalytic triad and Ser160 functions as a covalent nucleophile to the carbonyl carbon atom in the scissile ester bond, as in other carboxylesterases (Fig. 2a).”

Yes, but without any kinetic data or the direct observation of a covalent adduct, you are still assuming the enzyme proceeds as stated. Hence, rather than stating as a fact that “Ser160 functions as a covalent nucleophile...” this should be tempered with saying that Ser160 is “assumed” or “postulated” to function in this manner.

[RESPONSE] Thank you. We revised the sentence as you commented.

“At the active site of IsPETase, three residues Ser160, His237, and Asp206 form a catalytic triad and Ser160 is postulated to function as a covalent nucleophile to the carbonyl carbon atom in the scissile ester bond, as in other carboxylesterases (**Fig. 2a**).”

2. The structure of the enzyme is generally very similar to other Ser hydrolases and there is nothing presented that distinguishes it from these others. That it likely functions as a Ser

esterase could be readily predicted based on sequence alignments as searches. Unfortunately, this lessens the significance of this work beyond confirming what would have been predicted.

[RESPONSE] The major significance of our work is understanding how PET substrate is accommodated to IsPETase with the distinct substrate binding site resulting in superior PET degrading activity of this enzyme. As described in “Structural comparison of IsPETase with other PET degrading enzymes” section, we showed that IsPETase has unique structural features, especially on the substrate binding site. Such finding (along with other findings on mechanisms) cannot be obtained by prediction based on sequence alignments only as in Yoshida et al’s work.

Yes, but again the complex was “predicted” based on in silico docking experiments and not directly observed. Again, these predicted findings have been overstated and the tone of the discussion should reflect the predictive nature of the findings.

[RESPONSE] According to your comment, we revised the manuscript thoroughly. For example, “The substrate binding site is simulated to form a long, shallow L-shaped cleft on a flat surface” and “Met161 and Ile208 are also predicted to assist the binding of the first MHET by providing a hydrophobic surface”.

3. Line 123: The authors tried to both soak and co-crystalize with a substrate mimic but failed to obtain the structure of a complex. They assumed poor affinity of binding. Was this tested for in binding assays? If not, why not? Are there other compounds available that could have been tested? Unfortunately, without this, the authors resorted to molecular docking using a substrate mimic. Why this mimic?

[RESPONSE] To obtain the crystal structure of IsPETase with substrate, we used only BHET because there are no commercially available chemicals except BHET. Although we tried to measure the kinetic data of IsPETase using BHET, we failed to obtain reliable data due to the low solubility of BHET. Because we are not sure whether the affinity for BHET is low or not, we revised the sentence as follows: “...potentially because we could not use high concentration of BHET in cocrystallization and soaking due to its low solubility.” During this revision, we performed new docking calculation with longer substrate as suggested by the reviewers. At the end, we were able to make the docking complex of 4 MHET moieties and IsPETase.

OK - but again affinity of BHET could be readily determined by a ligand binding study using, eg., SPR or ITC.

[RESPONSE] As we mentioned in the first revision process, we could not measure the kinetics of the protein due to low solubility of BHET. We also attempted to measure the binding affinity using ITC. However, we could not obtain reasonable or publishable quality of data, so we could not present the data in this version of the manuscript.

4. The authors then proceed to describe the binding of this mimic in great detail using language that does not convey the hypothetical nature of the observations; the text reads as if this is so. This needs to be corrected.

[RESPONSE] Thank you. We revised the “Active site of *Is*PETase” section according to the reviewer’s advice.

[RESPONSE] Done.

5. Lines 198-211: Based on this docking experiment, the authors then present a mechanism of action involving the catalytic triad. They present only the first half of the reaction, that involving the formation of the first transition state. Water is not mentioned, which would be involved in releasing the second product through a second transition state. Having said this, the mechanism would involve a covalent intermediate, and likely a ping-pong, bi-bi pathway. Is there any evidence for either? Certainly the latter can be obtained quite readily through a kinetic characterization.

[RESPONSE] Since the catalytic mechanism involving the catalytic triad is already well known, we did not describe the catalytic process in detail (Reviewer 4 suggested to shorten the section). It is also well known that hydrolase enzymes have ping-pong bi-bi mechanism (Ringborg et al., (2016) React. Chem. Eng. 1: 10-22). As we responded to comment 2, *Is*PETase has similar catalytic residues and overall enzyme fold to other hydrolases. Therefore, we conclude that *Is*PETase also has ping-pong bi-bi mechanism.

I understand these assumptions - but they are nonetheless assumptions. Certainly as suggested by Reviewer 4, the discussion of the mechanism could be abbreviated, but given the broader readership of the Journal, it is not appropriate to ignore the second half of the reaction involving the addition of water to complete the hydrolytic reaction. This still needs to be included.

[RESPONSE] During the first revision process, we deleted the description on catalytic mechanism of the protein, because the other reviewers suggested that the mechanism is quite well known and does not need to describe in the manuscript. We also agree that the detailed

catalytic mechanism is important part of science. But, we have other more important findings to present in this paper and we think that the main focus of this work is not on the detailed catalytic mechanism. According to other reviewers' suggestions, we decided not to describe catalytic mechanism in detail.

6. Site-directed mutagenesis was performed to replace potentially important residues in order to predict their function. While it is unlikely that each of these replacements caused any folding issues, the authors should nonetheless have conducted an analysis to assure the reader, eg. Circular dichroism.

[RESPONSE] As the reviewer advised, we additionally performed CD experiments, and added the data to Supplementary Fig. 6. The CD spectra indicate that there are no significant differences among the variants and the wild-type IsPETase.

Thank you.

[RESPONSE] Thank you too.

In general, despite the claim on line 90 that the detailed mechanism of the esterase is presented, unfortunately without much direct observation and kinetic data, the mechanism can only be proposed at this time. Likewise, the process for PET film degradation described in lines 237-245 was not demonstrated experimentally and so the text is only speculative.

[RESPONSE] We agree that PET degradation process still remains speculative. It is extremely difficult to obtain the kinetic data of PET hydrolysis. However, in structural perspective, we can narrow down the possible positioning of the substrate to propose the mechanism. Moreover, in the revision, we present newly constructed variant (Arg280Ala) showing enhanced PET degrading activity. This successful enzyme engineering supports the reliability of docking calculation and identification of substrate binding site.

Nonetheless, my original concern still holds regarding the relatively strong language used to describe the mechanism; you still can only speculate or propose.

[RESPONSE] We revised our manuscript as responded above.

Minor points:

Line 41: I think the authors meant “not” instead of “now”

[RESPONSE] Thank you. Corrected. Thank you

Line 62: Only hydrolysis? Or do the authors mean enzyme activity, where hydrolysis is one

reaction type?

[RESPONSE] We changed “plastics” to “plastics with ester bond”. **Thank you**

Lines 156, 160, 167, 186 (and elsewhere?): “mutated” should read “replaced with” (as genes are mutated while amino acids are replaced).

[RESPONSE] We changed “mutated” to “replaced with”. **Thank you**

Lines 157, 164, 168, 170, 173, 182, 189 (and elsewhere?): “variants” should be used to replace “mutants”

[RESPONSE] We changed all “mutants” to “variants”. **Thank you**

Lines 187: “replacing” instead of “mutating”

[RESPONSE] We changed “mutating” to “replacing”. **Thank you**

Line 191: the H bond is “predicted” (it was not observed)

[RESPONSE] The H bond was removed in the revised manuscript. **OK**

Line 220: Nothing was truly observed, the authors are predicting or proposing.

[RESPONSE] Corrected as suggested.

As noted above several times, the tone of the discussion is still too strong and it needs to reflect the predictive nature of the observations that are based on theoretical/in silico studies.

[RESPONSE] Thank you. We revised our manuscript as responded above.

Reviewer #4 (Remarks to the Author):

The revised manuscript has improved considerably, and adequately addresses the main criticisms of the referees. In addition new results are included in the manuscript (e.g. docking of longer substrate, R280A mutant with enhanced PETase activity) which allow for a more thorough analysis and interesting discussion of the molecular basis of the enhanced PETase activity of this enzyme, as compared to its close homologs. Judged from the information presented in the manuscript the docking has been carried out carefully, and the results are highly relevant. In my view the manuscript is suitable for publication in Nature Comm.

[RESPONSE] Thank you very much for your invaluable comments on our original manuscript, which improved our manuscript.